# Efficiently Computing Local Lipschitz Constants of Neural Networks via Bound Propagation

**Zhouxing Shi[1], Yihan Wang[1], Huan Zhang[2], Zico Kolter[2,3], Cho-Jui Hsieh[1]**

[1]University of California, Los Angeles   [2]Carnegie Mellon University   [3]Bosch Center for AI

zshi@cs.ucla.edu, yihanwang@cs.ucla.edu, huan@huan-zhang.com
zkolter@cs.cmu.edu, chohsieh@cs.ucla.edu

## Abstract

Lipschitz constants are connected to many properties of neural networks, such as robustness, fairness, and generalization. Existing methods for computing Lipschitz constants either produce relatively loose upper bounds or are limited to small networks. In this paper, we develop an efficient framework for computing the $\ell_\infty$ local Lipschitz constant of a neural network by tightly upper bounding the norm of Clarke Jacobian via linear bound propagation. We formulate the computation of local Lipschitz constants with a linear bound propagation process on a high-order backward graph induced by the chain rule of Clarke Jacobian. To enable linear bound propagation, we derive tight linear relaxations for specific nonlinearities in Clarke Jacobian. This formulate unifies existing ad-hoc approaches such as RecurJac, which can be seen as a special case of ours with weaker relaxations. The bound propagation framework also allows us to easily borrow the popular Branch-and-Bound (BaB) approach from neural network verification to further tighten Lipschitz constants. Experiments show that on tiny models, our method produces comparable bounds compared to exact methods that cannot scale to slightly larger models; on larger models, our method efficiently produces tighter results than existing relaxed or naive methods, and our method scales to much larger practical models that previous works could not handle. We also demonstrate an application on provable monotonicity analysis. Code is available at `https://github.com/shizhouxing/Local-Lipschitz-Constants`.

## 1   Introduction

Lipschitz constants are important for characterizing many properties of neural networks, including robustness [44, 19, 45, 59, 21], fairness [12, 25], generalization [3], and explanation [14]. Local Lipschitz constants, which only need to hold for a small local region, can more precisely characterize the local behavior of a network. Intuitively they characterize how fast the output of the network changes between any two input within the region.

It is challenging to exactly and efficiently compute local Lipschitz constants. Naive approaches such as computing the product of the induced norm for all layers cannot capture local information and typically produce vacuous bounds. To compute tight and exact local Lipschitz constants, Jordan & Dimakis [24] considered small ReLU networks (e.g., up to tens of neurons), by solving a mixed-integer programming (MIP) problem to bound the norm of Clarke's generalized Jacobian [8]. However, solving MIP is often too costly and cannot scale to slightly larger networks. On the other hand, RecurJac [59] (an improved version of Fast-Lip [51]) is a specialized recursive algorithm for bounding the Jacobian and computing local Lipschitz constants, which is also relatively efficient but the produced bounds are relatively loose. Moreover, most of the existing works on local Lipschitz constants only used small toy models and cannot feasibly handle larger practical networks.

36th Conference on Neural Information Processing Systems (NeurIPS 2022).

Recently, in the field of neural network verification, many methods are proposed to compute provable output bounds for neural networks [26, 6, 11]; especially, linear bound propagation methods [52, 48, 58, 41] are becoming very successful [49, 61] because they are scalable and can be efficiently accelerated on GPUs. These methods propagate the linear relationship between layers, where nonlinearities in networks are relaxed into linear bounds. In this work, we ask the question if we can borrow these successful techniques from neural network verification to scale up the computation of local Lipschitz constants to larger and more practical networks.

In this paper, we aim to efficiently compute relatively tight $\ell_\infty$ local Lipschitz constants for neural networks using the bound propagation framework. We formulate this problem as upper bounding the $\ell_\infty$ norm of the Clarke Jacobian, and we formulate the computation for the Clarke Jacobian from a chain rule and its norm as a *higher-order backward computational graph* augmented to the original forward graph of the network. On the augmented computational graph, we generalize linear bound propagation to bound the Clarke Jacobian, and we thereby reformulate the problem of computing local Lipschitz constants under a linear bound propagation framework. On the backward graph, applying Clarke gradients in the chain rule is nonlinear and requires a linear relaxation. It is essentially formed by a *group* of functions and is different from *single* activation functions in regular neural network verification. We propose a tight and closed-form linear relaxation for Clarke gradients with an optimality guarantee on the tightness, and thereby we efficiently bound the Clarke Jacobian with linear bound propagation. We also show that RecurJac is a special case under our formulation where loose interval bounds instead of tight linear relaxation are used for nontrivial cases. Our formulation also allows us to develop a scalable and flexible framework enhanced by progress from recent neural network verifiers using linear bound propagation. We demonstrate that we can further tighten our bounds by Branch-and-Bound when time budget allows, for a trade-off between tightness and time cost.

Experiments show that our method efficiently produces tightest $\ell_\infty$ local Lipschitz constants compared to other relaxed methods, and is much more efficient than the exact MIP method. Moreover, our method scales to much larger models including practical convolutional neural networks (CNN) that previous works could not handle. We also demonstrate an application of our method for provably analyzing the monotonicity of neural networks.

## 2   Related Work

For neural networks, a loose global Lipschitz constant (upper bound) can be computed by the product of layer-wise induced norms [44]. Based on this product, Virmaux & Scaman [46] considered the effect of activation and approximated upper bounds which, however, are not guaranteed; Gouk et al. [16] used a power method for convolutional layers. LipSDP [13] used semidefinite programming (SDP) to compute tighter bounds. These works all compute global Lipschitz constants which can be much looser than local Lipschitz constants as they have to hold even for distinct input points and they cannot characterize the local behavior of neural networks.

We focus on local Lipschitz constants in this paper. On local Lipschitz constants, LipMIP [24] used mixed integer programming (MIP) to compute exact results. LipOpt [15] used polynomial optimization but is limited to smooth activations not including the widely used ReLU. LipBaB [4] combined relatively loose interval bound propagation [20, 35, 17] with branch-and-bound to compute exact local Lipschitz constants. All of these methods cannot scale to relatively larger models due to their computational cost. FastLip [51] and its improved version RecurJac [59] used recursive procedures to bound the Jacobian, and ZLip [23] used zonotope abstract interpretation with a focus on generative models. While these methods are much more efficient, their bounds are relatively loose due to the use of strictly looser relaxations compared to ours. In contrast, we compute $\ell_\infty$ local Lipschitz constants efficiently while our bounds are tighter than existing relaxed methods. Besides, there are also several other works on Lipschitz constants under different settings: Avant & Morgansen [2] derived layer-wise analytical bounds but only on a simplified definition for Lipschitz constants, which does not cover Lipschitzness in the entire local region; and Laurel et al. [28] proposed a dual number abstraction method to bound the Clarke Jacobian but they focused on non-smooth perturbations that can be represented by a single scalar, while we consider high dimensional perturbations.

Lipschitz constants can also be used to train certifiably robust neural networks, by computing margins from Lipschitz constants [45, 31, 21] or enforcing 1-Lipschitzness [1, 32, 42, 55–57], These methods

are competitive compared to certified training by directly bounding the output [35, 17, 60, 39, 50], but they are beyond the scope of this paper as we focus on pre-trained neural networks.

## 3 Background

### 3.1 ReLU Network

Suppose $f(\mathbf{x})$ is a $K$-way neural network classifier given a $d$-dimensional input $\mathbf{x} \in \mathbb{R}^d$, and then $f(\mathbf{x}) \in \mathbb{R}^K$. For the simplicity of presentation, we mainly focus on feedforward ReLU networks, but our method can also be applied to general network architectures and activations as will be discussed in Section 4. Suppose the network has $n$ layers, it takes input $h_0(\mathbf{x}) = \mathbf{x}$ and then computes

$$\forall i \in [n],\ z_i(\mathbf{x}) = \mathbf{W}_i h_{i-1}(\mathbf{x}) + \mathbf{b}_i,$$

$$\forall i \in [n-1],\ h_i(\mathbf{x}) = \sigma(z_i(\mathbf{x})),\ h_n(\mathbf{x}) = z_n(\mathbf{x}),$$

where $i \in [n]$ means $1 \le i \le n$, $z_i(\mathbf{x})$ is the pre-activation output of the $i$-th linear layer with weight $\mathbf{W}_i$ and bias $\mathbf{b}_i$, and $h_i(\mathbf{x})$ is the output after activation $\sigma(\cdot)$. This formulation is compatible with CNNs since convolutional layers are also linear.

### 3.2 Lipschiz Constant

The $(\alpha, \beta)$-Lipschitz constant of a network $f(\mathbf{x})$ over an open set $\mathcal{X} \in \mathbb{R}^d$ is defined as:

$$L^{(\alpha,\beta)}(f, \mathcal{X}) = \sup_{\mathbf{x}_1, \mathbf{x}_2 \in \mathcal{X},\ \mathbf{x}_1 \ne \mathbf{x}_2} \frac{\|f(\mathbf{x}_1) - f(\mathbf{x}_2)\|_\beta}{\|\mathbf{x}_1 - \mathbf{x}_2\|_\alpha}. \tag{1}$$

If $f$ is smooth and $(\alpha, \beta)$-Lipschitz continuous over $\mathcal{X}$, the Lipschitz constant can be computed by upper bounding the norm of Jacobian, i.e., $L^{(\alpha,\beta)}(f, \mathcal{X}) = \sup_{\mathbf{x} \in \mathcal{X}} \|\nabla f(\mathbf{x})\|_{\alpha,\beta}$. Since neural networks with non-smooth ReLU activation are non-smooth functions, we consider Clarke Jacobian [8] instead, which is defined as the convex hull of $\lim_{i \to \infty} \nabla f(\mathbf{x}_i)$ for any sequence $\{\mathbf{x}_i\}_{i=1}^\infty$ such that every $f(\mathbf{x}_i)$ is differentiable at $\mathbf{x}_i$ respectively. We denote the Clarke Jacobian at $\mathbf{x}$ as $\partial f(\mathbf{x})$, and then

$$L^{(\alpha,\beta)}(f, \mathcal{X}) = \sup_{\mathbf{x} \in \mathcal{X},\ \mathbf{J}(\mathbf{x}) \in \partial f(\mathbf{x})} \|\mathbf{J}(\mathbf{x})\|_{\alpha,\beta}. \tag{2}$$

Global Lipschitz constant, which considers the supremum over $\mathcal{X} = \mathbb{R}^d$, needs to guarantee Eq. (1) even for distant $\mathbf{x}_1, \mathbf{x}_2$, and thus it can be loose and cannot capture the local behavior of the network. We focus on $\ell_\infty$ local Lipschitz constants, where $\mathcal{X} = B_\infty(\mathbf{x}_0, \epsilon) := \{\mathbf{x} : \|\mathbf{x} - \mathbf{x}_0\|_\infty \le \epsilon\}$ is a small $\ell_\infty$-ball with radius $\epsilon$ around $\mathbf{x}_0$, and we take $\alpha = \beta = \infty$ for the definition in Eq. (1). While exactly computing Eq. (2) is possible for very small networks [24, 4], for slightly larger networks, we only expect to compute guaranteed upper bounds for Eq. (1) and Eq. (2), and we aim to make the upper bounds as tight as possible with an acceptable computational cost.

### 3.3 Backward Linear Bound Propagation

To bound the Clarke Jacobian, we will use linear bound propagation which is originally used for certifiably bounding the output of neural networks in neural network verification. We adopt backward bound propagation [58, 52, 41, 53] which typically propagates the linear relationship between the output layer to be bounded and all the previous layers it depends on, in a backward manner. Suppose we want to compute certified bounds for output layer $h_n(\mathbf{x})$ w.r.t. all $\mathbf{x}$ from a small domain $B_\infty(\mathbf{x}_0, \epsilon)$. Starting from the output layer, we take $\mathbf{A}_n = \mathbf{I}$, and then $h_n(\mathbf{x}) = \mathbf{A}_n h_n(\mathbf{x})$. For every $i \in [n]$, suppose $h_i(\mathbf{x})$ can be bounded by linear functions w.r.t. $h_{i-1}(\mathbf{x})$ parameterized by $\underline{\mathbf{P}}_i, \overline{\mathbf{P}}_i, \underline{\mathbf{q}}_i, \overline{\mathbf{q}}_i$ (the bounds element-wisely hold):

$$\underline{\mathbf{P}}_i h_{i-1}(\mathbf{x}) + \underline{\mathbf{q}}_i \le h_i(\mathbf{x}) \le \overline{\mathbf{P}}_i h_{i-1}(\mathbf{x}) + \overline{\mathbf{q}}_i. \tag{3}$$

Then for every $i = n, n-1, \cdots, 1$ in order, $\mathbf{A}_i h_i(\mathbf{x})$ can be recursively bounded by substituting $h_i(\mathbf{x})$ with Eq. (3):

$$\mathbf{A}_i h_i(\mathbf{x}) \le \mathbf{A}_{i-1} h_{i-1}(\mathbf{x}) + \mathbf{c}_i, \quad \mathbf{A}_{i-1} = [\mathbf{A}_i]_+ \overline{\mathbf{P}}_i + [\mathbf{A}_i]_- \underline{\mathbf{P}}_i, \quad \mathbf{c}_i = [\mathbf{A}_i]_+ \overline{\mathbf{q}}_i + [\mathbf{A}_i]_- \underline{\mathbf{q}}_i, \tag{4}$$

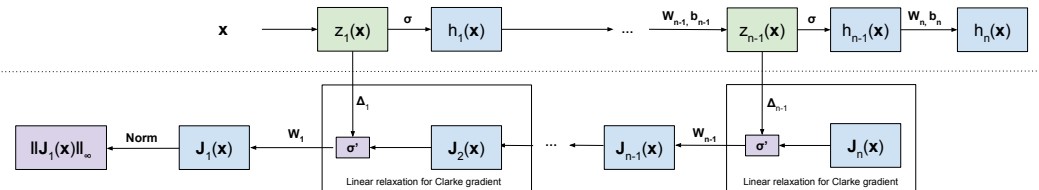

**Forward graph (original neural network computation)**

**Backward graph (norm of Clarke Jacobian)**

Figure 1: Illustration of our proposed framework where we formulate the computation for local Lipschitz constants by augmenting the original computation graph of the neural network (forward graph) with an additional subgraph for computing the norm of Clarke Jacobian (backward graph). The backward graph relies on the pre-activation values $(\mathbf{z}_1(\mathbf{x}), \cdots, \mathbf{z}_{n-1}(\mathbf{x}))$ in the forward graph to compute the Clarke gradient of activations. Linear relaxation for two categories of nonlinearities in the backward graph (nodes in purple), norm ($\|\mathbf{J}_1(\mathbf{x})\|_\infty$) and Clarke gradient ($\sigma'$), are detailed in Figure 5, Figure 2 and Figure 3. We also perform branch-and-bound (BaB) to tighten the bounds by branching pre-activation nodes in the forward graph (nodes in green).

where $[\cdot]_+$ stands for taking positive elements from the matrix or vector, and vice versa for $[\cdot]_-$. $\mathbf{A}_{i-1}$ stands for the coefficients of the linear relationship propagated from layer $i$ to layer $i-1$, and $\mathbf{c}_i$ is a bias term produced at this layer. By this recursive procedure in a backward manner and accumulating $\mathbf{c}_i (i \in [n])$, eventually bounds are propagated to the input layer as $h_n(\mathbf{x}) \le \mathbf{A}_0 \mathbf{x} + \sum_{i=1}^{n} \mathbf{c}_i$, which directly represents the linear relationship between the output layer and $\mathbf{x}$. We can eliminate $\mathbf{x}$ from $\mathbf{A}_0 \mathbf{x}$ to obtain the final bounds. When $\mathbf{x} \in B_\infty(\mathbf{x}_0, \epsilon)$, for the $j$-th dimension of $h_n(\mathbf{x})$, we can use the fact that $[\mathbf{A}_0]_{j,:} \mathbf{x} \le [\mathbf{A}_0]_{j,:} \mathbf{x}_0 + \epsilon \|[\mathbf{A}_0]_{j,:}\|_1$, where $[\mathbf{A}_0]_{j,:}$ stands for the $j$-th row in $\mathbf{A}_0$.

Parameters in Eq. (3) can be obtained by considering the relation between $h_i(\mathbf{x})$ and $h_{i-1}(\mathbf{x})$. We can first obtain bounds $\mathbf{l}_i \le z_i(\mathbf{x}) \le \mathbf{u}_i$ using bound propagation by viewing $z_i(\mathbf{x})$ as the output layer. Then a linear relaxation bounds activation $h_i(\mathbf{x}) = \sigma(z_i(\mathbf{x}))$ with linear functions of $z_i(\mathbf{x})$:

$$\forall \mathbf{l}_i \le z_i(\mathbf{x}) \le \mathbf{u}_i, \quad \underline{\mathbf{s}}_i z_i(\mathbf{x}) + \underline{\mathbf{t}}_i \le \sigma(z_i(\mathbf{x})) \le \overline{\mathbf{s}}_i z_i(\mathbf{x}) + \overline{\mathbf{t}}_i. \tag{5}$$

Intuitively, for the $j$-th neuron, $[\underline{\mathbf{s}}_i]_j [z_i(\mathbf{x})]_j + [\underline{\mathbf{t}}_i]_j$ and $[\overline{\mathbf{s}}_i]_j [z_i(\mathbf{x})]_j + [\overline{\mathbf{t}}_i]_j$ are two lines bounding $\sigma([z_i(\mathbf{x})]_j)$ against input $[z_i(\mathbf{x})]_j$. We provide a detailed derivation and illustration for ReLU in Appendix A.1. Then to satisfy Eq. (3), we can take:

$$\underline{\mathbf{P}}_i = \mathrm{diag}(\underline{\mathbf{s}}_i) \mathbf{W}_i, \quad \overline{\mathbf{P}}_i = \mathrm{diag}(\overline{\mathbf{s}}_i) \mathbf{W}_i, \quad \underline{\mathbf{q}}_i = \mathrm{diag}(\underline{\mathbf{s}}_i) \mathbf{b}_i + \underline{\mathbf{t}}_i, \quad \overline{\mathbf{q}}_i = \mathrm{diag}(\overline{\mathbf{s}}_i) \mathbf{b}_i + \overline{\mathbf{t}}_i. \tag{6}$$

Interested readers can find more details of backward bound propagation in the literature [58, 49]. In this work, we generalize this linear bound propagation to bound the Clarke Jacobian.

## 4 Methodology

We present our method in this section. We first formulate the local Lipschitz constant computation as a higher-order backward computational graph, and use bound propagation to obtain the bounds of the Clarke Jacobian. We discuss our methods for tightly bounding nonlinearities in Clarke Jacobian, and we show that RecurJac [59] is a special case with looser interval bounds. Finally, we use Branch-and-Bound to further tighten the bounds.

### 4.1 Clarke Jacobian for ReLU Networks

In ReLU networks, the ReLU activation for a single neuron is defined as $\sigma(x) = \max\{x, 0\}$ which is non-smooth at $x=0$. We consider its Clarke gradient [8] denoted as $\partial\sigma(x)$: when $x<0$, $\partial\sigma(x)=\{0\}$; when $x>0$, $\partial\sigma(x)=\{1\}$; and when $x=0$, $\partial\sigma(x)=[0,1]$. For the $i$-th layer ($1 \le i < n$), we use $\partial\sigma([z_i(\mathbf{x})]_j)$ to denote the Clarke gradient of the $j$-th neuron and $\partial\sigma(z_i(\mathbf{x}))$ to denote the Clarke gradient for all neurons in the layer. And we use a diagonal matrix $\Delta_i(\mathbf{x}) \in \partial\sigma(z_i(\mathbf{x}))$ to denote a Clarke gradient of neurons in the $i$-th layer layer, where $[\Delta_i(\mathbf{x})]_{jj} \in \partial\sigma([z_i(\mathbf{x})]_j)$. Then a chain rule can be used to compute the Clarke Jacobian for the whole network [22, 24]. Denote $\mathbf{J}_i(\mathbf{x}) \in \frac{\partial f(\mathbf{x})}{\partial h_{i-1}(\mathbf{x})}$

as a Clarke Jacobian w.r.t. the $(i-1)$-th layer in the chain rule. We have $\mathbf{J}_n(\mathbf{x}) = \mathbf{W}_n$, and for all $i \in [n-1]$, we have

$$\mathbf{J}_i(\mathbf{x}) \in \frac{\partial f(\mathbf{x})}{\partial h_{i-1}(\mathbf{x})} \left\{ \mathbf{J}_{i+1}(\mathbf{x})\Delta_i(\mathbf{x})\mathbf{W}_i : \mathbf{J}_{i+1}(\mathbf{x}) \in \frac{\partial f(\mathbf{x})}{\partial h_i(\mathbf{x})}, \Delta_i(\mathbf{x}) \in \partial\sigma(z_i(\mathbf{x})) \right\}. \tag{7}$$

Finally $\mathbf{J}_1(\mathbf{x}) \in \partial f(\mathbf{x})$ is a Clarke Jacobian of the entire network [24]. Since $f(\mathbf{x}) \in \mathbb{R}^K$ and $x \in \mathbb{R}^d$, we have $\mathbf{J}_1(\mathbf{x}) \in \mathbb{R}^{K \times d}$. As we aim to compute the $\ell_\infty$ local Lipschitz constant, we upper bound $\|\mathbf{J}_1(\mathbf{x})\|_\infty$ $(\mathbf{J}_1(\mathbf{x}) \in \partial f(x))$ w.r.t. all $\mathbf{x} \in \mathcal{X}$.

Given the forward pass computation of the original network, we can formulate a computational graph for $\|\mathbf{J}_1(\mathbf{x})\|_\infty$ in a backward pass accordingly as shown in Figure 1. We augment the original forward graph with a backward graph, where the backward graph has a dependency on the pre-activation outputs of layers in the forward graph, due to $\Delta_1, \Delta_2, \cdots, \Delta_{n-1}$. We build the augmented graph by traversing the forward graph, similar to automatic differentiation for neural network training. The formulation with a computational graph allows our method to easily generalize to different network architectures given the existing bound propagation framework for general neural networks [53].

We use the bound propagation framework described in Section 3.3 to upper bound $\|\mathbf{J}_1(\mathbf{x})\|_\infty$. On the backward graph, when bounds are propagated to $\mathbf{J}_i(\mathbf{x})$, similar to $\mathbf{A}_i$ and $\mathbf{c}_i$ in Eq. (4), we use $\widetilde{\mathbf{A}}_i$ to denote the coefficient matrix and $\widetilde{\mathbf{c}}_i$ to denote the bias term for the backward graph, and then we have:

$$\forall i \in [n], \quad \|\mathbf{J}_1(\mathbf{x})\|_\infty \le \mathbf{J}_i(\mathbf{x})\widetilde{\mathbf{A}}_i + \sum_{j=0}^{i-1} \widetilde{\mathbf{c}}_j. \tag{8}$$

Note that we have $\widetilde{\mathbf{A}}_i$ on the right of $\mathbf{J}_i(\mathbf{x})$ rather than the left, since $\Delta_i(\mathbf{x})\mathbf{W}_i$ is multiplied on the right of $\mathbf{J}_{i+1}(\mathbf{x})$ in Eq. (7). As intermediate bounds for $z_i(\mathbf{x})$ are needed for bounding the forward graph, we also need intermediate bounds $\mathbf{L}_i \le \mathbf{J}_i(\mathbf{x}) \le \mathbf{U}_i$ $(\forall \mathbf{x} \in \mathcal{X})$ for the backward graph by starting bound propagation from $\mathbf{J}_i(\mathbf{x})$ instead of $\|\mathbf{J}_1(\mathbf{x})\|_\infty$. These intermediate bounds are used in the linear relaxation for nonlinearities in the backward graph. There are particularly two categories of nonlinearities to be tackled on the backward graph: 1) $\ell_\infty$ norm in $\|\mathbf{J}_1(\mathbf{x})\|_\infty$; 2) $\mathbf{J}_{i+1}(\mathbf{x})\Delta_i$ $(\forall i \in [n-1])$ in Eq. (7), which applies the Clarke gradient of ReLU on the later layer's Clarke Jacobian. We handle them in Section 4.2 and Section 4.3 respectively. Then bounds can be propagated starting at $\|\mathbf{J}_1(\mathbf{x})\|_\infty$, to $\mathbf{J}_1(\mathbf{x}), \mathbf{J}_2(\mathbf{x}), \cdots, \mathbf{J}_n(\mathbf{x})$ in order. Bounds are eventually propagated to $\mathbf{J}_n(\mathbf{x})$ as Eq. (8) with $i = n$, where $\mathbf{J}_n(\mathbf{x})$ can be substituted with $\mathbf{W}_n$, and thereby we obtain a final upper bound for $\|\mathbf{J}_1(\mathbf{x})\|_\infty$.

## 4.2 Norm of Clarke Jacobian

The $\ell_\infty$ norm of Clarke Jacobian is the first nonlinearity we would encounter during backward bound propagation, as it is the last node in the backward graph (see Figure 1). The norm is computed as $\|\mathbf{J}_1(\mathbf{x})\|_\infty = \max_{1 \le k \le K} \sum_{j=1}^d |[\mathbf{J}_1(\mathbf{x})]_{kj}|$, where $K$ rows in $\mathbf{J}_1(\mathbf{x})$ can be bounded separately, and after that we can aggregate results on $K$ rows by taking the $\max$. For the simplicity of the following analysis, we assume $K = 1$ as we can handle one row in $\mathbf{J}_1(\mathbf{x})$ at each time, but they can still be batched in implementation. Then Clarke Jacobian $\mathbf{J}_i(\mathbf{x})$ $(i \in [n])$ and its bounds $\mathbf{L}_i$, $\mathbf{U}_i$ can be viewed as row vectors, and we will use $[\cdot]_j$ to denote the $j$-th element.

By a tight linear relaxation for $|\cdot|$ as shown in Figure 5 in Appendix A.2 (only the upper bound is needed), we have the following bound for $\|\mathbf{J}_1(\mathbf{x})\|_\infty$:

**Proposition 1** (Linear relaxations for matrix $\|\cdot\|_\infty$ norm). *For all $x \in \mathcal{X}$, suppose $\forall j \in [d]$, $[\mathbf{L}_1]_j \le [\mathbf{J}_1(\mathbf{x})]_j \le [\mathbf{U}_1]_j$, and $\mathbf{J}_1(\mathbf{x})$ is a row vector, we have a bound for its matrix $\|\cdot\|_\infty$ norm:*

$$\|\mathbf{J}_1(\mathbf{x})\|_\infty \le \mathbf{J}_1(\mathbf{x})\widetilde{\mathbf{A}}_1 + \widetilde{\mathbf{c}}_0,$$

$$where \; [\widetilde{\mathbf{A}}_1]_j = \begin{cases} \dfrac{\big|[\mathbf{U}_1]_j\big| - \big|[\mathbf{L}_1]_j\big|}{[\mathbf{U}_1]_j - [\mathbf{L}_1]_j} & [\mathbf{L}_1]_j < [\mathbf{U}_1]_j \\ 0 & [\mathbf{L}_1]_j = [\mathbf{U}_1]_j \end{cases}, \quad [\widetilde{\mathbf{c}}_0]_j = -[\widetilde{\mathbf{A}}_1]_j[\mathbf{L}_1]_j + \big|[\mathbf{L}_1]_j\big|.$$

We prove it in Appendix D.1, and this propagates bounds from the norm of Clarke Jacobian to $\mathbf{J}_1(\mathbf{x})$.

## 4.3 Clarke Gradient of Activation Functions

For each layer $i \in [n-1]$ in the chain rule for computing the Clarke Jacobian as Eq. (7), in addition to the fixed weight matrix $\mathbf{W}_i$, there is a $\mathbf{J}_{i+1}(\mathbf{x})\Delta_i(\mathbf{x})$ term, where $\Delta_i(\mathbf{x})$ is determined by pre-activation value $z_i(\mathbf{x})$ from the forward graph. For the $j$-th neuron in the layer, $[\mathbf{J}_{i+1}(\mathbf{x})\Delta_i(\mathbf{x})]_j = [\mathbf{J}_{i+1}(\mathbf{x})]_j [\Delta_i(\mathbf{x})]_{jj}$. When $[\mathbf{u}_i]_j < 0$ or $[\mathbf{l}_i]_j > 0$, we have $[\Delta_i(\mathbf{x})]_{jj} = 0$ and $[\Delta_i(\mathbf{x})]_{jj} = 1$ respectively fixed, and then for these special cases $[\mathbf{J}_{i+1}(\mathbf{x})\Delta_i(\mathbf{x})]_j$ is already linear w.r.t. $[\mathbf{J}_{i+1}(\mathbf{x})]_j$, and thus a linear relaxation is not needed. Otherwise, $[\mathbf{J}_{i+1}(\mathbf{x})\Delta_i(\mathbf{x})]_j$ is nonlinear, since both $[\mathbf{J}_{i+1}(\mathbf{x})]_j$ and $[\Delta(\mathbf{x})]_{jj}$ may vary given different $\mathbf{x} \in \mathcal{X}$. We thereby relax the nonlinearity $[\mathbf{J}_{i+1}(\mathbf{x})]_j [\Delta_i(\mathbf{x})]_{jj}$ for $[\Delta_i(\mathbf{x})]_{jj} = [0, 1]$.

As illustrated in Figure 2, the solid blue line and solid red line are the exact upper and lower bound respectively for $[\mathbf{J}_{i+1}(\mathbf{x})\Delta_i(\mathbf{x})]_j$, which are piecewise linear w.r.t. $[\mathbf{J}_{i+1}(\mathbf{x})]_j$. When $[\mathbf{L}_{i+1}]_j \geq 0$ or $[\mathbf{U}_{i+1}]_j \leq 0$, $[\mathbf{J}_{i+1}(\mathbf{x})\Delta_i(\mathbf{x})]_j$ can be exactly bounded by linear functions w.r.t. $[\mathbf{J}_{i+1}(\mathbf{x})]_j$:

$$\begin{cases} 0 \leq [\mathbf{J}_{i+1}(\mathbf{x})\Delta_i(\mathbf{x})]_j \leq [\mathbf{J}_{i+1}(\mathbf{x})]_j & \text{when } [\mathbf{L}_{i+1}]_j \geq 0, [\Delta_i(\mathbf{x})]_{jj} = [0, 1], \\ [\mathbf{J}_{i+1}(\mathbf{x}) \leq [\mathbf{J}_{i+1}(\mathbf{x})\Delta_i(\mathbf{x})]_j \leq 0 & \text{when } [\mathbf{U}_{i+1}]_j \leq 0, [\Delta_i(\mathbf{x})]_{jj} = [0, 1]. \end{cases} \quad (9)$$

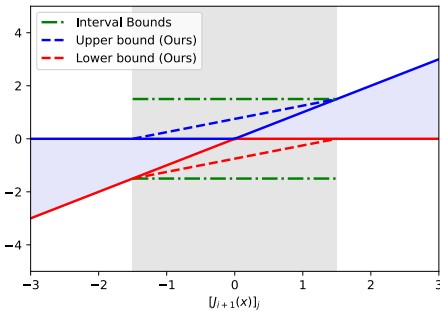

Figure 2: For a neuron $j$ in layer $i$, when $[\Delta_i(\mathbf{x})]_{jj} = [0, 1]$, we plot $[\mathbf{J}_{i+1}(\mathbf{x})\Delta_i(\mathbf{x})]_j$ as a group of functions w.r.t. $[\mathbf{J}_{i+1}(\mathbf{x})]_j$ (violet area). In this example, $-1.5 \leq [\mathbf{J}_{i+1}(\mathbf{x})]_j \leq 1.5$. Our linear relaxation (dashed blue and green lines) is much tighter than interval bounds (dashed green lines) by Zhang et al. [59].

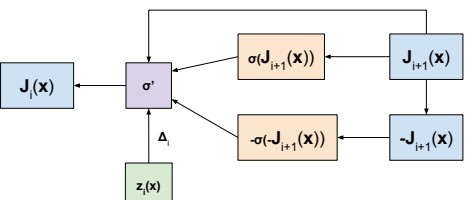

Figure 3: Computation graph for bounding each Clarke gradient. It is the internal computation for boxes in Figure 1. Two ReLU nodes are inserted as per Proposition 2. In bound propagation, the $\sigma'$ node takes intermediate bounds from $z_i(\mathbf{x})$ to decide whether bounds are directly propagated from $\mathbf{J}_{i+1}(\mathbf{x})$ (no relaxation needed), or indirectly via $\sigma(\mathbf{J}_{i+1}(\mathbf{x}))$ and $-\sigma(-\mathbf{J}_{i+1}(\mathbf{x}))$ (with relaxation), for each neuron respectively.

The most nontrivial case appears when $[\mathbf{L}_{i+1}]_j < 0 < [\mathbf{U}_{i+1}]_j$, where exact bounds for $[\mathbf{J}_{i+1}(\mathbf{x})\Delta_i(\mathbf{x})]_j$ are nonlinear given the input region. We relax the exact bounds into linear bounds that can be handled by linear bound propagation. We observe that the exact upper bound is essentially a ReLU function of $\mathbf{J}_{i+1}(\mathbf{x})$ (solid blue line in Figure 2), and the exact lower bound can be viewed as a ReLU function flipped both horizontally and vertically (solid red line in Figure 2). Formally, we have:

**Proposition 2** (Exact lower and and upper bounds for the Clarke gradient of a ReLU neuron). *For the $j$-th neuron in layer $i (i \in [n-1])$, if $[\Delta_i(\mathbf{x})]_{jj} = [0, 1]$, we have:*

$$[\mathbf{J}_{i+1}(\mathbf{x})\Delta_i(\mathbf{x})]_j \geq \min\{[\mathbf{J}_{i+1}(\mathbf{x})]_j, 0\}, \quad [\mathbf{J}_{i+1}(\mathbf{x})\Delta_i(\mathbf{x})]_j \leq \max\{[\mathbf{J}_{i+1}(\mathbf{x})]_j, 0\}, \quad (10)$$

*which can be rewritten using ReLU activation:*

$$\min\{[\mathbf{J}_{i+1}(\mathbf{x})]_j, 0\} = -\sigma(-[\mathbf{J}_{i+1}(\mathbf{x})]_j), \quad \max\{[\mathbf{J}_{i+1}(\mathbf{x})]_j, 0\} = \sigma([\mathbf{J}_{i+1}(\mathbf{x})]_j). \quad (11)$$

Its correctness can be easily verified by considering the sign of $[\mathbf{J}_{i+1}(\mathbf{x})]_j$ as shown in Appendix D.2. We then decompose $[\mathbf{J}_{i+1}(\mathbf{x})\Delta_i(\mathbf{x})]_j$ into two ReLU activations (Eq. (11)), as illustrated in Figure 3.

ReLU can be further relaxed by Eq. (5), as derived in Appendix D.3, which yields:

**Proposition 3** (Linear relaxations for Clarke Jacobian of a ReLU neuron). *For the $j$-th neuron in the $i$-th layer ($i \in [n-1]$), given $[\Delta_i(\mathbf{x})]_{jj} = [0, 1]$ and $[\mathbf{L}_{i+1}]_j \leq [\mathbf{J}_{i+1}(\mathbf{x})]_j \leq [\mathbf{U}_{i+1}]_j$, we have the following relaxation:*

$$[\mathbf{J}_{i+1}(\mathbf{x})\Delta_i(\mathbf{x})]_j \geq [\underline{\widetilde{\mathbf{s}}}_{i+1}]_j [\mathbf{J}_{i+1}(\mathbf{x})]_j + [\underline{\widetilde{\mathbf{t}}}_{i+1}]_j, \quad [\mathbf{J}_{i+1}(\mathbf{x})\Delta_i(\mathbf{x})]_j \leq [\overline{\widetilde{\mathbf{s}}}_{i+1}]_j [\mathbf{J}_{i+1}(\mathbf{x})]_j + [\overline{\widetilde{\mathbf{t}}}_{i+1}]_j,$$

$$\text{where} \quad [\widetilde{\underline{\mathbf{s}}}_{i+1}]_j = \frac{-\sigma(-[\mathbf{U}_{i+1}]_j) + \sigma(-[\mathbf{L}_{i+1}]_j)}{[\mathbf{U}_{i+1}]_j - [\mathbf{L}_{i+1}]_j}, \quad [\widetilde{\underline{\mathbf{t}}}_{i+1}]_j = -[\widetilde{\underline{\mathbf{s}}}_{i+1}]_j[\mathbf{L}_{i+1}]_j - \sigma(-[\mathbf{L}_{i+1}]_j),$$

$$[\widetilde{\overline{\mathbf{s}}}_{i+1}]_j = \frac{\sigma([\mathbf{U}_{i+1}]_j) - \sigma([\mathbf{L}_{i+1}]_j)}{[\mathbf{U}_{i+1}]_j - [\mathbf{L}_{i+1}]_j}, \quad [\widetilde{\overline{\mathbf{t}}}_{i+1}]_j = -[\widetilde{\overline{\mathbf{s}}}_{i+1}]_j[\mathbf{L}_{i+1}]_j + \sigma([\mathbf{L}_{i+1}]_j).$$

The linear relaxation in Proposition 3 corresponds to dashed blue and red lines shown in Figure 2 for the relaxed upper bound and lower bound respectively. And Eq. (9) is a special case for the relaxation in Proposition 3. In the following theorem, we show that relaxation in Proposition 3 is the tightest linear relaxation:

**Theorem 4** (Optimality of linear relaxations for the Clarke gradient of a ReLU neuron)**.** *For the $j$-th neuron in the $i$-th layer ($i \in [n-1]$), and any linear bound coefficients $\underline{s}, \underline{t}, \overline{s}, \overline{t}$ satisfying*

$$\forall J \in \left[ [\mathbf{L}_{i+1}]_j, [\mathbf{U}_{i+1}]_j \right], \forall [\Delta_i(\mathbf{x})]_{jj} \in [0,1], \quad \underline{s}J + \underline{t} \le J \cdot [\Delta_i(\mathbf{x})]_{jj} \le \overline{s}J + \overline{t}, \quad (12)$$

*we show that they produce looser relaxations:*

$$(\underline{s}, \underline{t}) \ne ([\widetilde{\underline{\mathbf{s}}}_{i+1}]_j, [\widetilde{\underline{\mathbf{t}}}_{i+1}]_j) \implies \forall [\mathbf{L}_{i+1}]_j < J < [\mathbf{U}_{i+1}]_j, \quad \underline{s}J + \underline{t} < [\widetilde{\underline{\mathbf{s}}}_{i+1}]_j J + [\widetilde{\underline{\mathbf{t}}}_{i+1}]_j,$$

$$(\overline{s}, \overline{t}) \ne ([\widetilde{\overline{\mathbf{s}}}_{i+1}]_j, [\widetilde{\overline{\mathbf{t}}}_{i+1}]_j) \implies \forall [\mathbf{L}_{i+1}]_j < J < [\mathbf{U}_{i+1}]_j, \quad \overline{s}J + \overline{t} > [\widetilde{\overline{\mathbf{s}}}_{i+1}]_j J + [\widetilde{\overline{\mathbf{t}}}_{i+1}]_j,$$

*where $[\widetilde{\underline{\mathbf{s}}}_{i+1}]_j, [\widetilde{\underline{\mathbf{t}}}_{i+1}]_j, [\widetilde{\overline{\mathbf{s}}}_{i+1}]_j, [\widetilde{\overline{\mathbf{t}}}_{i+1}]_j$ are defined in Proposition 3.*

This theorem, proved in Appendix D.4, states that our proposed linear relaxation is provably tighter than any other valid linear relaxation. While there are previous works tightening the linear relaxation for activation functions [58, 34, 54], we consider the optimal linear relaxation for a group of functions (as shown in Figure 2) instead of a single activation function, and our relaxation is closed-form with a provable optimality, without gradient-based optimization [34, 54].

### 4.4 Connections to RecurJac [59]

While RecurJac is a recursive algorithm specialized for local Lipschitz constants and it was not related to bound propagation in the original paper, we find that it is essentially a special case under our formulation with linear bound propagation, but it uses looser interval bound relaxation instead of our tight linear relaxation proposed in Proposition 2 and Proposition 3 (note that *interval* bound propagation is also a special case of *linear* bound propagation, but interval bounds use relaxation with zero slope only instead of general non-zero slopes). Figure 2 shows a nontrivial case where $[\mathbf{L}_{i+1}]_j < 0 < [\mathbf{U}_{i+1}]_j$ and $[\Delta_i(\mathbf{x})]_{jj} = [0,1]$. RecurJac takes interval bounds as the horizontal green dashed lines with a zero slope, and the interval bounds are looser than our linear relaxation with non-zero slopes, i.e., the gap between upper and lower bounds is larger. In Appendix E, we show that if we use interval relaxation when $[\mathbf{L}_{i+1}]_j < 0 < [\mathbf{U}_{i+1}]_j$, our framework will be equivalent to RecurJac. Compared to RecurJac, we not only have a more general formulation with linear bound propagation but also produce tighter results.

### 4.5 Branch and Bound

Formulating the backward computational graph also allows us to utilize recent progress in linear bound propagation for neural network verification, to further tighten the results. In neural network verification, Branch-and-Bound (BaB) has been used to compute tighter bounds [26, 6, 7, 47, 48, 54, 49] by branching activations or input and then bounding each smaller subdomain respectively. We also utilize BaB to tighten our bounds when time budget allows.

We denote $\mathcal{C}_0 = \{(\mathbf{l}_1, \mathbf{u}_1), (\mathbf{l}_2, \mathbf{u}_2), \cdots (\mathbf{l}_{n-1}, \mathbf{u}_{n-1})\}$ as the domain of all pre-activation bounds of ReLU neurons in the forward computational graphs before BaB. We use $\mathcal{C}$ to denote a pool of domains that we currently have, and initially $\mathcal{C} = \{\mathcal{C}_0\}$. BaB aims to recursively split domains in $\mathcal{C}$ into smaller subdomains $\mathcal{C}_1, \mathcal{C}_2, \cdots$, where $\mathcal{C}_0 = \mathcal{C}_1 \bigcup \mathcal{C}_2 \bigcup \cdots$, and bounds for each subdomain can be computed respectively using bound propagation, as linear relaxation can generally be tighter with smaller domains.

At each iteration, we take $B$ domains in $\mathcal{C}$ with loosest bounds, where $B$ is the batch size for BaB. For each domain, we choose a layer $\hat{i}$ and a ReLU neuron $\hat{j}$ in layer $\hat{i}$ on the forward graph, such that the

neuron has uncertain Clarke gradient ($[\mathbf{l}_{\hat{i}}]_{\hat{j}} \leq 0 \leq [\mathbf{u}_{\hat{i}}]_{\hat{j}}$). We branch this domain into two subdomains by branching ($[\mathbf{l}_{\hat{i}}]_{\hat{j}}, [\mathbf{u}_{\hat{i}}]_{\hat{j}}$) into ($[\mathbf{l}_{\hat{i}}]_{\hat{j}}, -\tilde{\epsilon}$) and ($\tilde{\epsilon}, [\mathbf{l}_{\hat{i}}]_{\hat{j}}$) respectively, where $\tilde{\epsilon}$ is a sufficiently small value (e.g., $10^{-9}$), so that the Clarke gradient for the branched neuron becomes certain in each subdomain (0 and 1 respectively) with other neurons unchanged. We compute the bounds for the new subdomains, and we repeat this process until reaching a time limit or there is no domain left to branch. Finally, the bound of the domain with the loosest bound in $\mathcal{C}$ is the tightened result.

We use a heuristic score to decide which neuron to branch for a given domain, by estimating the potential improvement to the bounds [6]. For branching a ReLU neuron on the forward graph, we estimate the potential improvement on the backward graph. Suppose neuron $\hat{j}$ in layer $\hat{i}$ is branched, we estimate the gap between the lower and upper bound in the liner relaxation, as the gap can be closed after the branching. We estimate the gap (the gap between the blue dashed line and the red dashed line in Figure 2) as $\frac{1}{2}([\mathbf{U}_{\hat{i}+1}]_{\hat{j}} - [\mathbf{L}_{\hat{i}+1}]_{\hat{j}})^2$, and we multiply it by coefficient $[\widetilde{\mathbf{A}}_{\hat{i}}]_{\hat{j}}$ from the bound propagation as the heuristic score. We thereby branch neurons with highest scores.

## 5 Experiments

In the experiments, we focus on evaluating the tightness of $\ell_\infty$ local Lipschitz constant bounds and the computational cost. There are three parts: 1) We compare the tightness and efficiency on relatively small models and synthetic data, to accommodate slower baselines; 2) We also use practical image datasets with larger models; 3) Finally, we demonstrate an application of our method on analyzing the monotonicity of neural networks. Additional experimental details are provided in Appendix C.

**Baselines** We compare our methods to the following baselines: **NaiveUB** multiplies the induced norm of each layer, which can scale to arbitrarily large models but the bound is often vacuous. **LipMIP** [24] computes exact and tightest local Lipschitz constants using MIP but can only work for tiny models. During the process of solving MIP, an upper bound of the result may be available and is gradually improved, and thus for models that LipMIP cannot finish within a time budget, we report the upper bound obtained at the timeout. **LipSDP** [13] computes upper bounds of $\ell_2$ local Lipschitz constants using SDP, and we convert their $\ell_2$ results into upper bounds for the $\ell_\infty$-case by a $\sqrt{d}$ factor. Since LipSDP by itself does not support the $\ell_\infty$ case which may not be the intended use of LipSDP, the results converted from the $\ell_2$ case are for reference only. **LipBaB** [4] uses loose interval bounds with branch-and-bound. **RecurJac** [59] is a recursive algorithm computing relaxed local Lipschitz constants which can be seen as a special case of ours with weaker relaxations. The baselines except NaiveUB do not support CNN in their implementation, and we convert the CNN models to equivalent MLP models for these baselines.

Table 1: Local Lipschitz constant values and runtime (seconds) on MLP and CNN models with growing width for a 16-dimensional synthetic input data point. Smaller values are tighter results. Width for MLP stands for number of neurons in each hidden layer, and width for CNN stands for number of filters in each convolutional layer. "C" and "F" in the model names for CNNs denote the number of convolutional layers and fully-connected layers respectively. We set a timeout of 1000s for LipMIP and LipSDP, and 60s for BaB. "*" denotes that we report the upper bound LipMIP returns at timeout, and "-" means LipSDP cannot return an upper bound at timeout.

| Method | 3-Layer MLP | | | | | | CNN-2C1F | | | | | |
| | Width=32 | | Width=64 | | Width=128 | | Width=4 | | Width=8 | | Width=16 | |
| | Value | Runtime | Value | Runtime | Value | Runtime | Value | Runtime | Value | Runtime | Value | Runtime |
|---|---|---|---|---|---|---|---|---|---|---|---|---|
| NaiveUB | 24.31 | 0.05 | 33.02 | 0.01 | 49.39 | 0.00 | 24.38 | 0.00 | 39.99 | 0.01 | 51.68 | 0.03 |
| LipMIP | 12.13 | 16.25 | 102.64 | 1,000.05* | 456.89 | 1,000.12* | 9.54 | 1,000.08* | 57.88 | 1,000.17* | 628.91 | 1,000.40* |
| LipSDP | 21.49 | 11.01 | 27.27 | 103.04 | - | - | 11.03 | 729.92 | - | - | - | - |
| LipBaB | 12.13 | 2.92 | 30.59 | 63.16 | 73.71 | 60.98 | 5.39 | 61.62 | 15.34 | 61.43 | 36.40 | 81.82 |
| RecurJac | 12.38 | 17.40 | 20.25 | 17.07 | 47.23 | 16.66 | 5.02 | 16.21 | 9.02 | 16.73 | 38.19 | 16.38 |
| Ours (w/o BaB) | 12.28 | 6.95 | 17.45 | 6.42 | 35.66 | 6.67 | 4.69 | 7.29 | 8.41 | 7.40 | 30.28 | 7.29 |
| Ours (w/ BaB) | 12.13 | 8.13 | 16.30 | 13.57 | 28.69 | 60.10 | 4.69 | 7.59 | 8.19 | 52.29 | 28.6 | 60.1 |

### 5.1 Comparison on a Synthetic Dataset

We follow Jordan & Dimakis [24] and train several small models on a synthetic dataset. We compare different methods on models with varying width as shown in Table 1, and we also show results with varying depth and $\epsilon$ in Appendix B.1. Some of these models are already large for LipMIP and we set a time limit of 1000 seconds. For our BaB, we stop further branching after 60 seconds.

Without BaB, our method is very efficient while it already outperforms the baselines on tightness except for LipMIP on the smallest MLP. Compared to LipMIP, for the smallest MLP on which LipMIP does not timeout, our result without BaB is very close to but larger than their exact result. This smallest model serves as a sanity check to validate that our results should be no smaller than LipMIP's exact results. For other relatively larger models, LipMIP cannot produce exact results within the timeout and our result is much tighter than the upper bounds by LipMIP after 1000 seconds. And LipSDP fails to yield results within the timeout on many models. Although LipBaB has BaB to tighten the bounds, their results are still loose within the time budget, since using loose interval bounds in their BaB is inefficient. RecurJac is a relatively efficient baseline but still returns looser bounds compared to our method with tight linear relaxation. Our BaB further improves the bounds and consistently produces tightest results within a reasonable time budget.

## 5.2 Comparison on Image Datasets

Table 2: Average local Lipschitz constant values and runtime (seconds) on MNIST. The Lipschitz constants are evaluated on the first 100 examples in the test set. We set a timeout value of 120s for LipMIP and 60s for BaB. "*" denotes that we report the upper bound LipMIP returns at timeout, and "-" means LipMIP and LipBaB cannot return any valid upper bound at timeout on the CNN model.

| Method | 3-layer MLP | | CNN-2C2F | |
| | Value | Runtime | Value | Runtime |
| --- | --- | --- | --- | --- |
| NaiveUB | 3,257.16 | 0.00 | 80,239.62 | 0.00 |
| LipMIP | 14,218.99* | 120.51 | - | - |
| LipBaB | 947.69 | 62.77 | - | - |
| RecurJac | 1,091.31 | 0.22 | 12,514.55 | 115.43 |
| Ours (w/o BaB) | 688.15 | 4.95 | 5,473.03 | 8.21 |
| Ours | 397.25 | 52.23 | 5,458.84 | 60.04 |

Table 3: Average local Lipschitz constant values and runtime (seconds) on CIFAR-10 and TinyImageNet. The Lipschitz constants are evaluated on the first 100 examples in the test set. RecurJac's original implementation cannot handle large models here, and its results are obtained by supporting its relaxation in our implementation.

| Dataset | CIFAR-10 | | TinyImageNet | |
| Model | CNN-2C2F | CNN-4C2F | CNN-2C2F | CNN-4C2F |
| --- | --- | --- | --- | --- |
| NaiveUB | 1707252.88 | 365293440.00 | 1512185.25 | 87148664.00 |
| RecurJac | 79275.44 | 12502332.00 | 24031.58 | 714189.12 |
| Value (ours) | 18638.14 | 1049447.88 | 4556.61 | 25096.26 |
| Runtime (ours) | 11.17 | 29.79 | 95.42 | 137.04 |

Table 4: Percentage of examples on which the predicted confidence for high income level is monotonically increasing (↑) or decreasing (↓) w.r.t. each feature, by Recurjac and our method respectively.

| Method | Monotonicity | Age | Education num | Capital gain | Capital loss | Hours-per-week |
| --- | --- | --- | --- | --- | --- | --- |
| RecurJac | ↑ | 32% | 55% | 0% | 4% | 95% |
| | ↓ | 0% | 0% | 0% | 0% | 0% |
| Ours | ↑ | 40% | 58% | 5% | 7% | 98% |
| | ↓ | 0% | 0% | 0% | 0% | 0% |

To further evaluate the scalability of different methods to relatively large models, we then conduct experiments on image datasets including MNIST [30], CIFAR-10 [27], and TinyImagenet [29]. We use a 3-layer MLP and a 4-layer CNN on MNIST, and we use 4-layer and 6-layer CNN models on CIFAR-10 and TinyImageNet. All the models here are too large for LipSDP. LipMIP and LipBaB can only handle the MLP model on MNIST, and they cannot return any valid upper bound for the CNN models within a reasonable time budget. RecurJac can handle the CNN model on MNIST, but other CNN models on CIFAR-10 and TinyImageNet are still too large for RecurJac after converted to equivalent MLP models. In contrast, our formulation with linear bound propagation allows us to utilize the efficient bound propagation for convolutional layers from Xu et al. [53]. Since RecurJac is a special case under our formulation, we implement RecurJac's loose relaxation to obtain results for RecurJac on CNN models.

We show results on MNIST in Table 2. Our results are much tighter compared to the baselines even if we do not enable BaB, and BaB can further tighten results. On CIFAR-10 and TinyImageNet, we

do not enable BaB on to save time cost on relatively larger models, and we show results in Table 3. Our method can scale to these larger CNNs that previous works could not handle. And our results are much tighter than NaiveUB and also RecurJac reimplemented in bound propagation. We present additional results on models with different random initialization in Appendix B.2.

### 5.3 An Application on Monotonicity Analysis

We further use an application to demonstrate that our method can be used to verify the monotonicity of neural networks. Given a network, we aim to verify whether the network's output is a monotonic function w.r.t. each input feature. Monotonicity can be preferred properties in some real-world scenarios [40, 9, 33, 43]. In this experiment, we adopt the Adult dataset [5], with a binary classification task about predicting income level. There are only several continuous features, and we aim to check the monotonicity of income level w.r.t. age, education level, capital gain, capital loss, and hours per week. We train an MLP model and verify its monotonicity on the first 100 test examples. Signs of Clarke Jacobian can be used to check monotonicity [37], and we achieve it by checking the bounds of Clarke Jacobian. For each continuous feature $j$, we set an input domain $\mathcal{X}_j$ where only this particular feature can be varied between the minimum and maximum possible values in the dataset, while we keep other features fixed. And then we obtain bounds on the Clarke Jacobian of the output class for high income level, as $\mathbf{L}_1 \leq \mathbf{J}_1(\mathbf{x}) \leq \mathbf{U}_1$ ($\forall \mathbf{x} \in \mathcal{X}_j$). Then if $[\mathbf{L}_1]_j > 0$, the predicted confidence on high income level is monotonically increasing w.r.t. feature $j$, and vice versa. We count the percentage of examples that satisfy each type of monotonicity respectively. We show results in Table 4. The predicted confidence on high income level is monotonically increasing w.r.t. age, education number, and hours per week on at least part of the examples, which is reasonable. On the other hand, the models learn little monotonicity on capital gain or loss features. Compared to RecurJac, our method verifies more monotonicity. We expect the gap to be larger when the models and datasets are larger.

## 6 Conclusion

In this paper, we propose an efficient framework to compute tight $\ell_\infty$ local Lipschitz constants for neural networks by upper bounding the norm of Clarke Jacobian, and we formulate the problem with linear bound propagation. We model the computation for the Clarke Jacobian as a backward computational graph and conduct linear bound propagation on the backward graph. We propose tight linear relaxation for nonlinearities in Clarke Jacobian with guarantees on the optimality. And we further tighten bounds with branch-and-bound when time budget allows. Experiments show that our method efficiently produces much tighter results compared to existing relaxed methods and can scale to larger convolutional networks on which previous works cannot handle. We also use our method to analyze the monotonicity of neural networks as a potential application.

**Limitations**   There are several limitations in this work and challenges for future works. It is still difficult for the proposed method to scale to deeper neural networks such as ResNet [18] with tens of convolutional layers on ImageNet [10]. It is also a common challenge in neural network verification, where efficiency and the tightness of certified bounds can be very limited on deep models. Besides, we have only focused on $\ell_\infty$ local Lipschitz constants in this paper but not other norms. Linear bound propagation computes certified bounds for each neuron respectively and thus aligns better with $\ell_\infty$ norm. Bounds may become significantly looser if it is applied to other norms such as $\ell_2$ norm. Previous state-of-the-art works [24, 59, 13] on local Lipschitz constants did not consider both $\ell_\infty$ norm and $\ell_2$ norm simultaneously either. Handling other norms is a challenge for future works. Moreover, in addition to verifying pre-trained models and computing local Lipschitz constants, we have not utilized the proposed method to train neural networks with stronger certified guarantees on Lipschitz constants or robustness, and it remains challenging for future works to better align verification and training.

## Funding Disclosure

This work is supported in part by NSF under IIS-2008173, IIS-2048280 and by Army Research Laboratory under W911NF-20-2-0158. Huan Zhang is supported by a grant from the Bosch Center for Artificial Intelligence.

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
