# A Additional Explanations

## A.1 Linear Relaxation for ReLU

Recall that as introduced in Section 3.3, given bounds of $z_i(\mathbf{x})$ as $\mathbf{l}_i \leq z_i(\mathbf{x}) \leq \mathbf{u}_i$, we aim to relax an activation $\sigma(z_i(\mathbf{x}))$ as

$$\forall \mathbf{l}_i \leq z_i(\mathbf{x}) \leq \mathbf{u}_i, \ \underline{\mathbf{s}}_i z_i(\mathbf{x}) + \underline{\mathbf{t}}_i \leq \sigma(z_i(\mathbf{x})) \leq \bar{\mathbf{s}}_i z_i(\mathbf{x}) + \bar{\mathbf{t}}_i.$$

For every neuron $j$, if $[\mathbf{l}_i]_j \geq 0$ or $[\mathbf{u}_i]_j \leq 0$, $\sigma(z_i(\mathbf{x})) = z_i(\mathbf{x})$ or $\sigma(z_i(\mathbf{x})) = 0$ respectively is already linear, and then we can simply take

$$[\underline{\mathbf{s}}_i]_j [z_i(\mathbf{x})]_j + [\underline{\mathbf{t}}_i]_j = [\bar{\mathbf{s}}_i]_j [z_i(\mathbf{x})]_j + [\bar{\mathbf{t}}_i]_j = \sigma(z_i(\mathbf{x})).$$

Otherwise when $[\mathbf{l}_i]_j < 0 < [\mathbf{u}_i]_j$, the upper bound can be the line passing ReLU activation at $[z_i(\mathbf{x})]_j = [\mathbf{l}_i]_j$ and $[z_i(\mathbf{x})]_j = [\mathbf{u}_i]_j$ respectively, i.e.,

$$[\bar{\mathbf{s}}_i]_j [z_i(\mathbf{x})]_j + [\bar{\mathbf{t}}_i]_j = \left(\frac{\sigma([\mathbf{u}_i]_j) - \sigma([\mathbf{l}_i]_j)}{[\mathbf{u}_i]_j - [\mathbf{l}_i]_j}\right)(z_i(\mathbf{x}) - [\mathbf{l}_i]_j) + \sigma([\mathbf{l}_i]_j).$$

The lower bound can be any line with a slope between 0 and 1 and a bias of 0, i.e., $0 \leq [\underline{\mathbf{s}}_i]_j \leq 1$, $[\underline{\mathbf{t}}_i]_j = 0$, where $[\underline{\mathbf{s}}_i]_j$ can be viewed as a parameter optimized with an objective of tightening output bounds [34, 54, 36]. We illustrate the linear relaxation in Figure 4.

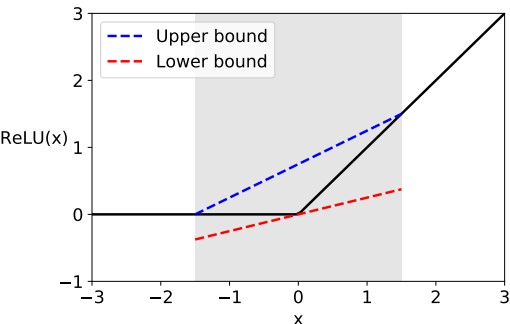

Figure 4: Two lines can bound the ReLU activation when the sign of its input is unstable ($-1.5 \leq x \leq 1.5$ in this example) and are used for bound propagation with linear relaxation.

## A.2 Linear Relaxation for Absolute Value Function

In Figure 5, we illustrate the linear relaxation for the absolute value function.

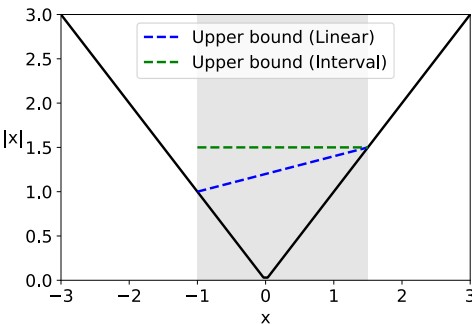

Figure 5: Given the bound on the input to an absolute value function ($-1 \leq x \leq 1.5$ in this example), a line can upper bound the function. We take the blue one with slope $\frac{1}{3}$, while Zhang et al. [59] would take the green one with slope 0, which is a looser upper bound.

## A.3 Applicability to Other Activations

Although we focus on ReLU networks and $\ell_\infty$ local Lipschitz constants in this paper, our method can also be applied to networks with other activations. When the activation is not ReLU, in Proposition 2, we can still obtain the range of $[\Delta_i(\mathbf{x})]_{jj}$ given pre-activation bounds $[\mathbf{l}_i]_j$ and $[\mathbf{u}_i]_j$. Suppose the Clarke gradient satisfies $[\Delta_i(\mathbf{x})]_{jj} \subseteq [l, u]$. Then

$$[\mathbf{J}_{i+1}(\mathbf{x})\Delta_i(\mathbf{x})]_j \leq \max\{[\mathbf{J}_{i+1}(\mathbf{x})]_j \cdot l, [\mathbf{J}_{i+1}(\mathbf{x})]_j \cdot u\} \coloneqq \tilde\sigma([\mathbf{J}_{i+1}(\mathbf{x})]_j),$$

and

$$
\begin{aligned}
& [\mathbf{J}_{i+1}(\mathbf{x})\Delta_i(\mathbf{x})]_j \\
\geq\ & \min\{[\mathbf{J}_{i+1}(\mathbf{x})]_j \cdot l, [\mathbf{J}_{i+1}(\mathbf{x})]_j \cdot u\} \\
=\ & -\max\{-[\mathbf{J}_{i+1}(\mathbf{x})]_j \cdot l, -[\mathbf{J}_{i+1}(\mathbf{x})]_j \cdot u\} \\
=\ & -\tilde\sigma(-[\mathbf{J}_{i+1}(\mathbf{x})]_j),
\end{aligned}
$$

where $\tilde\sigma(\cdot)$ is a Leaky-ReLU-like function. Thereby, in Proposition 3, we can replace ReLU $\sigma(\cdot)$ with $\tilde\sigma(\cdot)$ instead, and the proposition will hold for non-ReLU activations in the network.

# B Additional Experiments

## B.1 Additional Experiments on Synthetic Data

Table 5: Local Lipschitz constant values and runtime (seconds) on MLP models with varying depth on synthetic data. Settings other than the models are the same as those for experiments in Table 1.

| Method | MLP with width=32 | | | | | |
| | Depth=2 | | Depth=4 | | Depth=8 | |
| | Value | Runtime | Value | Runtime | Value | Runtime |
|---|---|---|---|---|---|---|
| NaiveUB | 29.25 | 0.00 | 36.99 | 0.00 | 180.87 | 0.00 |
| LipMIP | 15.42 | 6.78 | 12.32 | 381.62 | 18,938.35 | 1,000.11* |
| LipSDP | 25.88 | 9.97 | 24.57 | 28.55 | 49.00 | 1,826.92 |
| LipBaB | 15.42 | 5.87 | 13.74 | 60.10 | 963.83 | 60.67 |
| RecurJac | 15.45 | 17.19 | 13.95 | 23.64 | 195.16 | 42.09 |
| Ours (w/o BaB) | 16.00 | 2.27 | 13.24 | 9.02 | 35.64 | 22.11 |
| Ours (w/ BaB) | 15.42 | 4.89 | 12.36 | 15.92 | 30.32 | 60.03 |

Table 6: Local Lipschitz constant values and runtime (seconds) on an 3-layer MLP model with a width of 64, for input domains with varying radii $\epsilon$. Settings other than the model and $\epsilon$ are the same as those for experiments in Table 1.

| Method | 3-layer MLP with width=64 | | | | | | | |
| | $\epsilon=0.01$ | | $\epsilon=0.05$ | | $\epsilon=0.1$ | | $\epsilon=0.2$ | |
| | Value | Runtime | Value | Runtime | Value | Runtime | Value | Runtime |
|---|---|---|---|---|---|---|---|---|
| NaiveUB | 33.02 | 0.01 | 33.02 | 0.01 | 33.02 | 0.01 | 33.02 | 0.01 |
| LipMIP | 15.49 | 7.75 | 15.71 | 403.93 | 102.64 | 1,000.05* | 244.47 | 1,000.05* |
| LipSDP | 27.27 | 93.35 | 27.27 | 93.35 | 27.27 | 93.35 | 27.27 | 93.35 |
| LipBaB | 15.49 | 2.66 | 17.13 | 60.44 | 30.59 | 63.16 | 56.03 | 60.26 |
| RecurJac | 15.49 | 19.27 | 16.17 | 18.39 | 20.25 | 17.07 | 65.28 | 18.09 |
| Ours (w/o BaB) | 15.49 | 4.32 | 16.00 | 5.41 | 17.45 | 6.42 | 40.34 | 5.81 |
| Ours (w/ BaB) | 15.49 | 5.46 | 15.82 | 13.54 | 16.30 | 13.57 | 32.70 | 60.10 |

In this section, we show additional empirical results on the synthetic dataset. Settings are mostly similar to those for experiments in Section 5.1. In Table 5, we show results on models with varying depth. For deeper models, our method outperforms previous works with larger gaps. And in Table 6, we show results on varying radii $\epsilon$ for the input domain. Our method outperforms previous methods for computing local Lipschitz constants (baselines except for NaiveUB and LipSDP) with larger gaps when the input domain has larger radii. But since local Lipschitz constants aim to analyze the properties of the network within a small local region, the input domain should not be too large, otherwise it is essentially no longer local.

## B.2 Repeated Experiments on MNIST

On MNIST, we show experiments where we train each of 3-layer MLP and CNN-2C2F with 5 different random initialization respectively, and other settings remain the same as those in Table 2. We compute local Lipschitz constants for these models, and we report the mean and standard deviations for the 5 runs in each setting respectively. We show the results in Table 7. Our improvement over the baselines is significant.

Table 7: Results on models with 5 different random initialization on MNIST. Other settings remain similar as those in Table 2.

| Method | 3-layer MLP | | CNN-2C2F | |
| | Value | Runtime | Value | Runtime |
| --- | --- | --- | --- | --- |
| NaiveUB | $3218.17 \pm 289.99$ | $0.00 \pm 0.00$ | $92108.52 \pm 13962.82$ | $0.00 \pm 0.00$ |
| LipMIP | $14053.27 \pm 264.03*$ | $120.29 \pm 0.11$ | - | - |
| LipBaB | $988.66 \pm 86.93$ | $63.01 \pm 0.63$ | - | - |
| RecurJac | $1153.68 \pm 97.87$ | $0.38 \pm 0.08$ | $11174.06 \pm 1342.49$ | $114.18 \pm 3.40$ |
| Ours (w/o BaB) | $\mathbf{734.40 \pm 63.69}$ | $4.26 \pm 0.37$ | $\mathbf{4895.50 \pm 517.39}$ | $7.89 \pm 0.20$ |
| Ours | $\mathbf{423.30 \pm 38.25}$ | $58.54 \pm 3.15$ | $\mathbf{4878.96 \pm 518.49}$ | $60.05 \pm 0.01$ |

# C Experiment Details

## C.1 Synthetic Datasets

For experiments using synthetic data, we generate the datasets following LipMIP [24] which generates a randomized classification dataset given the input dimension and number of classes. We use 10 classes. We generate a dataset for MLP models and CNN models respectively, where the input dimension is 16 for MLP models, and $1 \times 8 \times 8$ for CNN models. For each dataset, we generate 7000 examples for training and 3000 examples for testing.

## C.2 Model Structures

**Synthetic Data** For experiments on synthetic data, the numbers of hidden neurons in MLP models and the number of convolution filters are listed as "width" in the tables for the corresponding results. The numbers of layers are also mentioned in the tables. All the convolution kernels have a size of $3 \times 3$, a stride of 1, and no padding. Each CNN model only has one fully-connected output layer for classification, and there is no hidden fully-connected layer.

**MNIST** For experiments on MNIST, each hidden layer in the 3-layer MLP model has 20 neurons; and for the CNN-2C2F model, there are 8 convolution filters in each of the two convolutional layers, where each convolutional filter has a kernel size of $4 \times 4$, a stride of 1, and no padding, and there is a fully-connected hidden layer with 100 neurons.

**CIFAR-10** For experiments on CIFAR-10, the CNN-2C2F model has 32 convolution filters in each of the two convolutional laeyrs, where each convolutional filter has a kernel size of $3 \times 3$, a stride of 1, and no padding, and there is a fully-connected hidden layer with 256 neurons; and the CNN-4C2F model has two additional convolutional layers with same hyperparameters.

**TinyImageNet** For experiments on TinyImageNet, the CNN-2C2F model has 32 convolution filters in each of the two convolutional laeyrs, where each convolutional filter has a kernel size of $3 \times 3$, a stride of 2, and a padding size of 1, and there is a fully-connected hidden layer with 256 neurons; and the CNN-4C2F model has two additional convolutional layers with same hyperparameters.

**Monotonicity Analysis** For the monotonicity analysis, we use a 4-layer MLP model where each hidden layer has 512 neurons.

## C.3 Model Training

We use the Adam optimizer to train the models. For experiments on the synthetic data, we train each model for 10 epochs with a learning rate of $10^{-3}$. For experiments on the image datasets, we train

each model for 30 epochs with a learning rate of $5 \times 10^{-4}$. For experiments on the monotonicity analysis, we train the model for 10 epochs with a learning rate of $5 \times 10^{-4}$. Hyperparameters were not specifically tuned, as the focus on this paper is not on training models.

### C.4  Compute Resources

All experiments are done on internal GPU servers, and each experiment only uses one single GPU. We use a NVIDIA RTX A6000 GPU for experiments on CIFAR-10 and TinyImageNet, and we use a NVIDIA GeForce RTX 2080 Ti GPU for experiments on other datasets.

### C.5  Existing Assets

The implementation is partly based on the following open-source code repositories under BSD-3-Clause license:

- `auto_LiRPA` (https://github.com/Verified-Intelligence/auto_LiRPA);
- `alpha-beta-CROWN`
  (https://github.com/Verified-Intelligence/alpha-beta-CROWN).

The following datasets are used, with no license found:

- MNIST (http://yann.lecun.com/exdb/mnist);
- CIFAR-10 (https://www.cs.toronto.edu/~kriz/cifar.html);
- TinyImageNet (http://cs231n.stanford.edu/tiny-imagenet-200.zip);
- Adult dataset (https://archive.ics.uci.edu/ml/datasets/adult).

These resources are publicly available. We believe the data do not contain personally identifiable information or offensive content.

## D  Proofs for Our Linear Relaxation

### D.1  Proof for Proposition 1

*Proof.* For every $j \in [d]$, we have $[\mathbf{L}_1]_j \leq [\mathbf{J}_1(\mathbf{x})]_j \leq [\mathbf{U}_1]_j$ ($\forall \mathbf{x} \in \mathcal{X}$). In the special case with $[\mathbf{L}_1]_j = [\mathbf{U}_1]_j = [\mathbf{J}_1(\mathbf{x})]_j$, we take

$$[\widetilde{\mathbf{A}}_1]_j = 0, \quad [\widetilde{\mathbf{c}}_0]_j = -[\widetilde{\mathbf{A}}_1]_j[\mathbf{L}_1]_j + \big|[\mathbf{L}_1]_j\big| = \big|[\mathbf{J}_1(\mathbf{x})]_j\big|,$$

and thereby $|[\mathbf{J}_1(\mathbf{x})]_j| = [\widetilde{\mathbf{A}}_1]_j[\mathbf{J}_1(\mathbf{x})]_j + [\widetilde{\mathbf{c}}_0]_j$.

Otherwise, assuming $[\mathbf{L}_1]_j < [\mathbf{U}_1]_j$, we have $0 \leq \frac{[\mathbf{J}_1(\mathbf{x})]_j - [\mathbf{L}_1]_j}{[\mathbf{U}_1]_j - [\mathbf{L}_1]_j} \leq 1$. Meanwhile, since $|\cdot|$ is a convex function, we have

$$\forall 0 \leq t \leq 1, \quad |t[\mathbf{U}_1]_j + (1-t)[\mathbf{L}_1]_j| \leq t|[\mathbf{U}_1]_j| + (1-t)|[\mathbf{L}_1]_j|. \tag{13}$$

By taking $t = \frac{[\mathbf{J}_1(\mathbf{x})]_j - [\mathbf{L}_1]_j}{[\mathbf{U}_1]_j - [\mathbf{L}_1]_j}$, for Eq. (13), the left-hand-side becomes

$$|t[\mathbf{U}_1]_j + (1-t)[\mathbf{L}_1]_j| = |[\mathbf{J}_1(\mathbf{x})]_j|,$$

and the right-hand-side becomes

$$t|[\mathbf{U}_1]_j| + (1-t)|[\mathbf{L}_1]_j| = \frac{(|[\mathbf{U}_1]_j| - |[\mathbf{L}_1]_j|) \cdot [\mathbf{J}_1(\mathbf{x})]_j - [\mathbf{L}_1]_j \cdot |[\mathbf{U}_1]_j| + |[\mathbf{L}_1]_j| \cdot [\mathbf{U}_1]_j}{[\mathbf{U}_1]_j - [\mathbf{L}_1]_j},$$

and thereby we have

$$|[\mathbf{J}_1(\mathbf{x})]_j| \leq \frac{(|[\mathbf{U}_1]_j| - |[\mathbf{L}_1]_j|) \cdot [\mathbf{J}_1(\mathbf{x})]_j - [\mathbf{L}_1]_j \cdot |[\mathbf{U}_1]_j| + |[\mathbf{L}_1]_j| \cdot [\mathbf{U}_1]_j}{[\mathbf{U}_1]_j - [\mathbf{L}_1]_j}.$$

By taking $[\widetilde{\mathbf{A}}_1]_j = \frac{|[\mathbf{U}_1]_j| - |[\mathbf{L}_1]_j|}{[\mathbf{U}_1]_j - [\mathbf{L}_1]_j}$, and $[\widetilde{\mathbf{c}}_0]_j = -[\widetilde{\mathbf{A}}_1]_j[\mathbf{L}_1]_j + |[\mathbf{L}_1]_j|$, the right-hand-side can be simplified into $[\widetilde{\mathbf{A}}_1]_j[\mathbf{J}_1(\mathbf{x})]_j + [\widetilde{\mathbf{c}}_0]_j$, and thereby $|[\mathbf{J}_1(\mathbf{x})]_j| \leq [\widetilde{\mathbf{A}}_1]_j[\mathbf{J}_1(\mathbf{x})]_j + [\widetilde{\mathbf{c}}_0]_j$.

So far, we have $|[\mathbf{J}_1(\mathbf{x})]_j| \leq [\widetilde{\mathbf{A}}_1]_j[\mathbf{J}_1(\mathbf{x})]_j + [\widetilde{\mathbf{c}}_0]_j$ hold for all $j \in [d]$. In Section 4.2, we have assumed that we handle one row in the Jacobian at each time, and $\mathbf{J}_1(\mathbf{x}) \in \mathbb{R}^{1 \times d}$ can be viewed as a row vector. We upper bound its norm as

$$\|\mathbf{J}_1(\mathbf{x})\|_\infty = \sum_{j=1}^{d} |[\mathbf{J}_1(\mathbf{x})]_j| \leq \sum_{j=1}^{d} ([\widetilde{\mathbf{A}}_1]_j[\mathbf{J}_1(\mathbf{x})]_j + [\widetilde{\mathbf{c}}_0]_j) = \widetilde{\mathbf{A}}_1\mathbf{J}_1(\mathbf{x}) + \widetilde{\mathbf{c}}_0.$$

$\square$

## D.2 Proof for Proposition 2

*Proof.* For the $j$-th neuron in layer $i(i \in [n-1])$, given $[\Delta_i(\mathbf{x})]_{jj} = [0, 1]$, if $[\mathbf{J}_{i+1}(\mathbf{x})]_j \geq 0$, we have

$$[\mathbf{J}_{i+1}(\mathbf{x})\Delta_i(\mathbf{x})]_j = [\mathbf{J}_{i+1}(\mathbf{x})]_j[\Delta_i(\mathbf{x})]_{jj} \geq 0,$$

and if $[\mathbf{J}_{i+1}(\mathbf{x})]_j < 0$, we have

$$[\mathbf{J}_{i+1}(\mathbf{x})\Delta_i(\mathbf{x})]_j = [\mathbf{J}_{i+1}(\mathbf{x})]_j[\Delta_i(\mathbf{x})]_{jj} \geq [\mathbf{J}_{i+1}(\mathbf{x})]_j,$$

and thus

$$[\mathbf{J}_{i+1}(\mathbf{x})\Delta_i(\mathbf{x})]_j \geq \min\{[\mathbf{J}_{i+1}(\mathbf{x})]_j, 0\}.$$

Similarly, we can also obtain $[\mathbf{J}_{i+1}(\mathbf{x})\Delta_i(\mathbf{x})]_j \leq \max\{[\mathbf{J}_{i+1}(\mathbf{x})]_j, 0\}$. Recall that ReLU activation is $\sigma(x) = \max\{x, 0\}$, and thus $\max\{[\mathbf{J}_{i+1}(\mathbf{x})]_j, 0\} = \sigma([\mathbf{J}_{i+1}(\mathbf{x})]_j)$, and $\min\{[\mathbf{J}_{i+1}(\mathbf{x})]_j, 0\} = -\max\{-[\mathbf{J}_{i+1}(\mathbf{x})]_j, 0\} = -\sigma(-[\mathbf{J}_{i+1}(\mathbf{x})]_j)$. $\square$

## D.3 Proof for Proposition 3

*Proof.* For the $j$-th neuron in layer $i(i \in [n-1])$, given $[\mathbf{L}_{i+1}]_j \leq [\mathbf{J}_{i+1}(\mathbf{x})]_j \leq [\mathbf{U}_{i+1}]_j$, we have $-[\mathbf{U}_{i+1}]_j \leq -[\mathbf{J}_{i+1}(\mathbf{x})]_j \leq -[\mathbf{L}_{i+1}]_j$. And by the convex relaxation of ReLU activation in many previous works [52, 41, 58], we have

$$\sigma([\mathbf{J}_{i+1}(\mathbf{x})]_j) \leq \frac{\sigma([\mathbf{U}_{i+1}]_j) - \sigma([\mathbf{L}_{i+1}]_j)}{[\mathbf{U}_{i+1}]_j - [\mathbf{L}_{i+1}]_j}([\mathbf{J}_{i+1}(\mathbf{x})]_j - [\mathbf{L}_{i+1}]_j) + \sigma([\mathbf{L}_{i+1}]_j),$$

$$\sigma(-[\mathbf{J}_{i+1}(\mathbf{x})]_j) \leq \frac{\sigma(-[\mathbf{L}_{i+1}]_j) - \sigma(-[\mathbf{U}_{i+1}]_j)}{[\mathbf{L}_{i+1}]_j - [\mathbf{U}_{i+1}]_j}([\mathbf{J}_{i+1}(\mathbf{x})]_j + [\mathbf{L}_{i+1}]_j) + \sigma(-[\mathbf{L}_{i+1}]_j),$$

and

$$-\sigma(-[\mathbf{J}_{i+1}(\mathbf{x})]_j) \geq -\frac{\sigma(-[\mathbf{L}_{i+1}]_j) - \sigma(-[\mathbf{U}_{i+1}]_j)}{[\mathbf{L}_{i+1}]_j - [\mathbf{U}_{i+1}]_j}([\mathbf{J}_{i+1}(\mathbf{x})]_j + [\mathbf{L}_{i+1}]_j) - \sigma(-[\mathbf{L}_{i+1}]_j).$$

By setting

$$[\underline{\widetilde{\mathbf{s}}}_{i+1}]_j = \frac{-\sigma(-[\mathbf{U}_{i+1}]_j) + \sigma(-[\mathbf{L}_{i+1}]_j)}{[\mathbf{U}_{i+1}]_j - [\mathbf{L}_{i+1}]_j}, \quad [\underline{\widetilde{\mathbf{t}}}_{i+1}]_j = -[\underline{\widetilde{\mathbf{s}}}_{i+1}]_j[\mathbf{L}_{i+1}]_j - \sigma(-[\mathbf{L}_{i+1}]_j),$$

$$[\overline{\widetilde{\mathbf{s}}}_{i+1}]_j = \frac{\sigma([\mathbf{U}_{i+1}]_j) - \sigma([\mathbf{L}_{i+1}]_j)}{[\mathbf{U}_{i+1}]_j - [\mathbf{L}_{i+1}]_j}, \quad [\overline{\widetilde{\mathbf{t}}}_{i+1}]_j = -[\overline{\widetilde{\mathbf{s}}}_{i+1}]_j[\mathbf{L}_{i+1}]_j + \sigma([\mathbf{L}_{i+1}]_j),$$

we have

$$-\sigma(-[\mathbf{J}_{i+1}(\mathbf{x})]_j) \geq [\underline{\widetilde{\mathbf{s}}}_{i+1}]_j[\mathbf{J}_{i+1}(\mathbf{x})]_j + [\underline{\widetilde{\mathbf{t}}}_{i+1}]_j.$$

$$\sigma([\mathbf{J}_{i+1}(\mathbf{x})]_j) \leq [\overline{\widetilde{\mathbf{s}}}_{i+1}]_j[\mathbf{J}_{i+1}(\mathbf{x})]_j + [\overline{\widetilde{\mathbf{t}}}_{i+1}]_j.$$

By further combining results from Proposition 2, we have

$$[\mathbf{J}_{i+1}(\mathbf{x})\Delta_i(\mathbf{x})]_j \geq -\sigma(-[\mathbf{J}_{i+1}(\mathbf{x})]_j) \geq [\underline{\widetilde{\mathbf{s}}}_{i+1}]_j[\mathbf{J}_{i+1}(\mathbf{x})]_j + [\underline{\widetilde{\mathbf{t}}}_{i+1}]_j,$$

$$[\mathbf{J}_{i+1}(\mathbf{x})\Delta_i(\mathbf{x})]_j \leq \sigma([\mathbf{J}_{i+1}(\mathbf{x})]_j) \leq [\overline{\widetilde{\mathbf{s}}}_{i+1}]_j[\mathbf{J}_{i+1}(\mathbf{x})]_j + [\overline{\widetilde{\mathbf{t}}}_{i+1}]_j.$$

$\square$

### D.4 Proof for Theorem 4

*Proof.* For the $j$-th neuron in the $i$-th layer ($i \in [n-1]$), a linear relaxation $\underline{s}J + \underline{t}$ is valid lower bound if and only if

$$\forall J \in \left[[\mathbf{L}_{i+1}]_j, [\mathbf{U}_{i+1}]_j\right], \forall [\Delta_i(\mathbf{x})]_{jj} \in [0,1], \;\; \underline{s}J + \underline{t} \leq J \cdot [\Delta_i(\mathbf{x})]_{jj}. \tag{14}$$

When $[\mathbf{U}_{i+1}]_j \geq [\mathbf{L}_{i+1}]_j \geq 0$, we have

$$[\widetilde{\underline{\mathbf{s}}}_{i+1}]_j = \frac{-\sigma(-[\mathbf{U}_{i+1}]_j) + \sigma(-[\mathbf{L}_{i+1}]_j)}{[\mathbf{U}_{i+1}]_j - [\mathbf{L}_{i+1}]_j} = \frac{0+0}{[\mathbf{U}_{i+1}]_j - [\mathbf{L}_{i+1}]_j} = 0,$$

$$[\widetilde{\underline{\mathbf{t}}}_{i+1}]_j = -[\widetilde{\underline{\mathbf{s}}}_{i+1}]_j[\mathbf{L}_{i+1}]_j - \sigma(-[\mathbf{L}_{i+1}]_j) = 0.$$

Then suppose there exists some $\hat{J}([\mathbf{L}_{i+1}]_j < \hat{J} < [\mathbf{U}_{i+1}]_j)$ such that $\underline{s}\hat{J} + \underline{t} > [\widetilde{\underline{\mathbf{s}}}_{i+1}]_j\hat{J} + [\widetilde{\underline{\mathbf{t}}}_{i+1}]_j$, we have $\underline{s}\hat{J} + \underline{t} > 0$. However, by taking $J = \hat{J}$ and $[\Delta_i(\mathbf{x})]_{jj} = 0$, $\underline{s}J + \underline{t} > 0$ while $J \cdot [\Delta_i(\mathbf{x})]_{jj} = 0$, and thus the condition $\underline{s}J + \underline{t} \leq J \cdot [\Delta_i(\mathbf{x})]_{jj} = 0$ is violated, and $\underline{s}J + \underline{t}$ cannot be a linear relaxation for the lower bound. And if $[\mathbf{L}_{i+1}]_j \leq [\mathbf{U}_{i+1}]_j \leq 0$, we have

$$[\widetilde{\underline{\mathbf{s}}}_{i+1}]_j = \frac{-\sigma(-[\mathbf{U}_{i+1}]_j) + \sigma(-[\mathbf{L}_{i+1}]_j)}{[\mathbf{U}_{i+1}]_j - [\mathbf{L}_{i+1}]_j} = \frac{-(-[\mathbf{U}_{i+1}]_j) + (-[\mathbf{L}_{i+1}]_j)}{[\mathbf{U}_{i+1}]_j - [\mathbf{L}_{i+1}]_j} = 1,$$

$$[\widetilde{\underline{\mathbf{t}}}_{i+1}]_j = -[\widetilde{\underline{\mathbf{s}}}_{i+1}]_j[\mathbf{L}_{i+1}]_j - \sigma(-[\mathbf{L}_{i+1}]_j) = 0.$$

Then suppose there exists some $\hat{J}([\mathbf{L}_{i+1}]_j < \hat{J} < [\mathbf{U}_{i+1}]_j)$ such that $\underline{s}\hat{J} + \underline{t} > [\widetilde{\underline{\mathbf{s}}}_{i+1}]_j\hat{J} + [\widetilde{\underline{\mathbf{t}}}_{i+1}]_j$, we have $\underline{s}\hat{J} + \underline{t} > \hat{J}$. However, by taking $J = \hat{J}$ and $[\Delta_i(\mathbf{x})]_{jj} = 1$, $\underline{s}J + \underline{t} > \hat{J}$ while $J \cdot [\Delta_i(\mathbf{x})]_{jj} = \hat{J}$, and thus the condition $\underline{s}J + \underline{t} \leq J \cdot [\Delta_i(\mathbf{x})]_{jj} = \hat{J}$ is violated, and $\underline{s}J + \underline{t}$ cannot be a linear relaxation for the lower bound.

Next, we consider the remaining case when $[\mathbf{L}_{i+1}]_j < 0 < [\mathbf{U}_{i+1}]_j$. We have

$$[\widetilde{\underline{\mathbf{s}}}_{i+1}]_j = \frac{-\sigma(-[\mathbf{U}_{i+1}]_j) + \sigma(-[\mathbf{L}_{i+1}]_j)}{[\mathbf{U}_{i+1}]_j - [\mathbf{L}_{i+1}]_j} = \frac{-[\mathbf{L}_{i+1}]_j}{[\mathbf{U}_{i+1}]_j - [\mathbf{L}_{i+1}]_j},$$

$$[\widetilde{\underline{\mathbf{t}}}_{i+1}]_j = -[\widetilde{\underline{\mathbf{s}}}_{i+1}]_j[\mathbf{L}_{i+1}]_j - \sigma(-[\mathbf{L}_{i+1}]_j) = \frac{[\mathbf{L}_{i+1}]_j[\mathbf{U}_{i+1}]_j}{[\mathbf{U}_{i+1}]_j - [\mathbf{L}_{i+1}]_j}.$$

Then suppose there exists some $\hat{J}([\mathbf{L}_{i+1}]_j < \hat{J} < [\mathbf{U}_{i+1}]_j)$ such that $\underline{s}\hat{J} + \underline{t} > [\widetilde{\underline{\mathbf{s}}}_{i+1}]_j\hat{J} + [\widetilde{\underline{\mathbf{t}}}_{i+1}]_j$, we denote $H = \underline{s}\hat{J} + \underline{t}$, and then $\underline{s}J + \underline{t} = \underline{s}(J - \hat{J}) + H$. We have

$$H > \frac{-[\mathbf{L}_{i+1}]_j}{[\mathbf{U}_{i+1}]_j - [\mathbf{L}_{i+1}]_j}\hat{J} + \frac{[\mathbf{L}_{i+1}]_j[\mathbf{U}_{i+1}]_j}{[\mathbf{U}_{i+1}]_j - [\mathbf{L}_{i+1}]_j},$$

and

$$H - [\mathbf{L}_{i+1}]_j = \frac{-[\mathbf{L}_{i+1}]_j(\hat{J} - [\mathbf{L}_{i+1}]_j)}{[\mathbf{U}_{i+1}]_j - [\mathbf{L}_{i+1}]_j} > 0 \implies H > [\mathbf{L}_{i+1}]_j.$$

According to conditions in Eq. (14), we have

$$\forall J \in \{[\mathbf{L}_{i+1}]_j, \hat{J}, [\mathbf{U}_{i+1}]_j\}, \;\; \forall [\Delta_i(\mathbf{x})]_{jj} \in [0,1], \;\; \underline{s}(J - \hat{J}) + H \leq J[\Delta_i(\mathbf{x})]_{jj}.$$

Then by taking $J = \hat{J}$ and $[\Delta_i(\mathbf{x})]_{jj} = 0$, we have $H \leq 0$; by taking $J = [\mathbf{L}_{i+1}]_j$ and $[\Delta_i(\mathbf{x})]_{jj} = 1$, we have

$$\underline{s}([\mathbf{L}_{i+1}]_j - \hat{J}) + H \leq [\mathbf{L}_{i+1}]_j,$$

and thus

$$\underline{s} \geq \frac{H - [\mathbf{L}_{i+1}]_j}{\hat{J} - [\mathbf{L}_{i+1}]_j} > 0.$$

And by taking $J = [\mathbf{U}_{i+1}]_j$, we have

$$
\begin{aligned}
&\underline{s}([\mathbf{U}_{i+1}]_j - \hat{J}) + H \\
\geq\ & \frac{H - [\mathbf{L}_{i+1}]_j}{\hat{J} - [\mathbf{L}_{i+1}]_j}([\mathbf{U}_{i+1}]_j - \hat{J}) + H \\
=\ & \frac{-[\mathbf{L}_{i+1}]_j}{[\mathbf{U}_{i+1}]_j - [\mathbf{L}_{i+1}]_j}([\mathbf{U}_{i+1}]_j - \hat{J}) + H \\
>\ & \frac{-[\mathbf{L}_{i+1}]_j}{[\mathbf{U}_{i+1}]_j - [\mathbf{L}_{i+1}]_j}([\mathbf{U}_{i+1}]_j - \hat{J}) + [\mathbf{L}_{i+1}]_j \\
=\ & \frac{[\mathbf{L}_{i+1}]_j(\hat{J} - [\mathbf{L}_{i+1}]_j)}{[\mathbf{U}_{i+1}]_j - [\mathbf{L}_{i+1}]_j} > 0,
\end{aligned}
$$

which violates Eq. (14) when $[\Delta_i(\mathbf{x})]_{jj} = 0$.

Thus, so far we have proved that if $\underline{s}J + \underline{t}$ is a valid linear relaxation for the lower bound, $\forall J\,([\mathbf{L}_{i+1}]_j < J < [\mathbf{U}_{i+1}]_j)$, $\underline{s}\hat{J} + \underline{t} \leq [\underline{\widetilde{\mathbf{s}}}_{i+1}]_j\hat{J} + [\underline{\widetilde{\mathbf{t}}}_{i+1}]_j$ must hold, i.e., it must be no tighter than the relaxation proposed by Proposition 3. And then we must have $\forall J\,([\mathbf{L}_{i+1}]_j < J < [\mathbf{U}_{i+1}]_j)$, $\underline{s}\hat{J} + \underline{t} < [\underline{\widetilde{\mathbf{s}}}_{i+1}]_j\hat{J} + [\underline{\widetilde{\mathbf{t}}}_{i+1}]_j$, unless $(\underline{s}, \underline{t}) = ([\underline{\widetilde{\mathbf{s}}}_{i+1}]_j, [\underline{\widetilde{\mathbf{t}}}_{i+1}]_j)$. And thereby,

$$(\underline{s}, \underline{t}) \neq ([\underline{\widetilde{\mathbf{s}}}_{i+1}]_j, [\underline{\widetilde{\mathbf{t}}}_{i+1}]_j) \implies \forall [\mathbf{L}_{i+1}]_j < J < [\mathbf{U}_{i+1}]_j,\ \underline{s}J + \underline{t} < [\underline{\widetilde{\mathbf{s}}}_{i+1}]_j J + [\underline{\widetilde{\mathbf{t}}}_{i+1}]_j.$$

Similarly, for the linear relaxation $\bar{s}J + \bar{t}$ on the upper bound satisfying,

$$\forall J \in \Big[[\mathbf{L}_{i+1}]_j, [\mathbf{U}_{i+1}]_j\Big], \forall [\Delta_i(\mathbf{x})]_{jj} \in [0,1],\ \ \bar{s}J + \bar{t} \geq J \cdot [\Delta_i(\mathbf{x})]_{jj},$$

we can also prove

$$(\bar{s}, \bar{t}) \neq ([\overline{\widetilde{\mathbf{s}}}_{i+1}]_j, [\overline{\widetilde{\mathbf{t}}}_{i+1}]_j) \implies \forall [\mathbf{L}_{i+1}]_j < J < [\mathbf{U}_{i+1}]_j,\ \bar{s}J + \bar{t} > [\overline{\widetilde{\mathbf{s}}}_{i+1}]_j J + [\overline{\widetilde{\mathbf{t}}}_{i+1}]_j.$$

$\square$

Thereby, we have shown the optimality of our linear relaxation, which is provably tighter than other relaxations such as interval bound-like relaxation in RecurJac [59]. Alternatively, if we directly adopt the relaxation for the multiplication of two variables proposed in Shi et al. [38] for the multiplication of $[\mathbf{J}_{i+1}(\mathbf{x})]_j$ and $[\Delta_i(\mathbf{x})]_{jj}$, it will produce a similarly loose relaxation as Zhang et al. [59].

## E    Connection with RecurJac

In this section, we explain in more detail that RecurJac [59] is a special case under our framework using linear bound propagation but RecurJac has used relatively loose interval bounds in non-trivial cases as partly illustrated in Figure 2. For the $j$-th neuron in the $i$-th layer, in cases where $[\mathbf{L}_{i+1}]_j < 0 < [\mathbf{U}_{i+1}]_j$ but excluding the situation where

$$i = n - 1,\ \ [\Delta_i]_{jj} = 0 \text{ or } [\Delta_i]_{jj} = 1,$$

if we use following interval relaxation instead of our relaxation proposed in Section 4.3,

$$[\mathbf{L}_{i+1}]_j[\mathbf{u}'_i]_j \leq [\mathbf{J}_{i+1}(\mathbf{x})\Delta_i(\mathbf{x})]_j \leq [\mathbf{U}_{i+1}]_j[\mathbf{u}'_i]_j \quad \text{for } [\mathbf{L}_{i+1}]_j < 0 < [\mathbf{U}_{i+1}]_j, \tag{15}$$

then our framework without BaB computes equivalent results as RecurJac given same bounds on $\Delta_i(\mathbf{x})$. This is an interval relaxation because the relaxed lower bound and upper bound no longer depend on $\mathbf{J}_{i+1}(\mathbf{x})$ and thereby constitute an interval, instead of linear functions w.r.t. $\mathbf{J}_{i+1}(\mathbf{x})$. In particular, for the case illustrated in Figure 2 with $[\Delta_i]_{jj} = [0, 1]$, Eq. (15) is equivalent to

$$[\mathbf{L}_{i+1}]_j \leq [\mathbf{J}_{i+1}(\mathbf{x})\Delta_i(\mathbf{x})]_j \leq [\mathbf{U}_{i+1}]_j \quad \text{for } [\mathbf{L}_{i+1}]_j < 0 < [\mathbf{U}_{i+1}]_j.$$

But the interval relaxation is also used in more trivial cases with $[\Delta_i(\mathbf{x})]_{jj} = 0$ or $[\Delta_i(\mathbf{x})]_{jj} = 1$ fixed except $i = n - 1$, where $[\mathbf{J}_{i+1}(\mathbf{x})\Delta_i(\mathbf{x})]_j$ is naturally a linear function of $\mathbf{J}_{i+1}(\mathbf{x})$ and can allow linear bound propagation to continue and propagate the bounds to $\mathbf{J}_{i+1}(\mathbf{x})$. Using interval bounds also loosen the bounds in these cases.

Next, we compare with "Algorithm 1" in Zhang et al. [59] and show the equivalence. For each layer $i \in [n]$, we use $\hat{\mathbf{L}}_i$ and $\hat{\mathbf{U}}_i$ to denote the lower bound and upper bound on the Clarke Jacobian $\mathbf{J}_i(\mathbf{x})$ computed by our framework with the alternative relaxation in Eq. (15). And we use $\tilde{\mathbf{L}}_i$ and $\tilde{\mathbf{U}}_i$ to denote bounds computed by RecurJac, i.e., "$\mathbf{L}^{(-l)}$" and "$\mathbf{U}^{(-l)}$" in Zhang et al. [59] where $l = i$. We aim to show that $\hat{\mathbf{L}}_i = \tilde{\mathbf{L}}_i$ and $\hat{\mathbf{U}} = \tilde{\mathbf{U}}_i$ hold for all $i \in [n]$.

**Equivalence on the last layer**  First, as mentioned in Section 4.1, we have $\mathbf{J}_n(\mathbf{x}) = \mathbf{W}_n$ which does not depend on any relaxation, and thus $\hat{\mathbf{L}}_n = \hat{\mathbf{U}}_n = \mathbf{J}_n(\mathbf{x}) = \mathbf{W}_n$ remains unchanged even if we change to use RecurJac's relaxation. Meanwhile, RecurJac also returns $\tilde{\mathbf{L}}_n = \tilde{\mathbf{U}}_n = \mathbf{W}_n$ (in Zhang et al. [59]'s "Algorithm 1", it is showed that "$\mathbf{L}^{(-l)} = \mathbf{U}^{(-l)} = \mathbf{W}^{(l)}$" when "$l = H$", where $H$ is used to denote the number of layers in Zhang et al. [59] equivalent to $n$ here).

**Equivalence on the second last layer**  Second, following Eq. (7), we have $\mathbf{J}_{n-1}(\mathbf{x}) = \mathbf{J}_n(\mathbf{x})\Delta_{n-1}(\mathbf{x})\mathbf{W}_{n-1}$. Note that the interval relaxation in Eq. (15) is not used for layer $n-1$. For every neuron $j$, if $[\Delta_i(\mathbf{x})]_{jj} = 0$ or $[\Delta_i(\mathbf{x})]_{jj} = 1$ is fixed, linear bound propagation is straightforward by merging $[\Delta_{n-1}(\mathbf{x})]_{jj}$ and $[\mathbf{W}_{n-1}]_{j,:}$ into new linear coefficients:

$$[\mathbf{J}_n(\mathbf{x})]_j[\Delta_{n-1}(\mathbf{x})]_{jj}[\mathbf{W}_{n-1}]_{j,:} = [\mathbf{J}_n(\mathbf{x})]_j([\Delta_{n-1}(\mathbf{x})]_{jj}[\mathbf{W}_{n-1}]_{j,:}).$$

For other cases with $[\Delta_i(\mathbf{x})]_{jj} = [0, 1]$, note that since $\mathbf{J}_n(\mathbf{x}) = \hat{\mathbf{L}}_n = \hat{\mathbf{U}}_n = \mathbf{W}_n$ is constant, this is a special case and at least one of $[\hat{\mathbf{L}}_n]_j = [\mathbf{W}_n]_j \geq 0$ and $[\hat{\mathbf{U}}_n]_j = [\mathbf{W}_n]_j \leq 0$ holds. Then the relaxation in Eq. (9) is used. Following Eq. (4), when using bound propagation to bound $\mathbf{J}_{n-1}(\mathbf{x})$, we have

$$
\begin{aligned}
& \mathbf{J}_{n-1}(\mathbf{x}) \\
= {} & \mathbf{J}_n(\mathbf{x})\Delta_{n-1}(\mathbf{x})\mathbf{W}_{n-1} \\
\geq {} & \sum_{\substack{[\Delta_{n-1}(\mathbf{x})]_{jj}=0 \\ \text{or } [\Delta_{n-1}(\mathbf{x})]_{jj}=1}} [\mathbf{J}_n(\mathbf{x})]_j([\Delta_{n-1}(\mathbf{x})]_{jj}[\mathbf{W}_{n-1}]_{j,:}) \\
& + \sum_{\substack{[\Delta_{n-1}(\mathbf{x})]_{jj}=[0,1], \\ [\mathbf{L}_n]_j \geq 0}} [\mathbf{J}_n(\mathbf{x})]_j[\mathbf{W}_{n-1}]_{j,:,-} + \sum_{\substack{[\Delta_{n-1}(\mathbf{x})]_{jj}=[0,1], \\ [\mathbf{U}_n]_j \leq 0}} [\mathbf{J}_n(\mathbf{x})]_j[\mathbf{W}_{n-1}]_{j,:,+} \\
= {} & \sum_{\substack{[\Delta_{n-1}(\mathbf{x})]_{jj}=0 \\ \text{or } [\Delta_{n-1}(\mathbf{x})]_{jj}=1}} [\mathbf{W}_n]_j([\Delta_{n-1}(\mathbf{x})]_{jj}[\mathbf{W}_{n-1}]_{j,:}) \\
& + \sum_{\substack{[\Delta_{n-1}(\mathbf{x})]_{jj}=[0,1], \\ [\mathbf{L}_n]_j \geq 0}} [\mathbf{W}_n]_j[\mathbf{W}_{n-1}]_{j,:,-} + \sum_{\substack{[\Delta_{n-1}(\mathbf{x})]_{jj}=[0,1], \\ [\mathbf{U}_n]_j \leq 0}} [\mathbf{W}_n]_j[\mathbf{W}_{n-1}]_{j,:,+}.
\end{aligned}
$$

It is easy to verify that the above bound is equivalent to

$$\mathbf{J}_{n-1}(\mathbf{x}) \geq ([\mathbf{W}_n]_+\mathbf{l}_{n-1} + [\mathbf{W}_n]_-\mathbf{u}_{n-1})[\mathbf{W}_{n-1}]_+ + ([\mathbf{W}_n]_+\mathbf{u}_{n-1} + [\mathbf{W}_n]_-\mathbf{l}_{n-1})[\mathbf{W}_{n-1}]_-.$$

Similarly, we also have

$$\mathbf{J}_{n-1}(\mathbf{x}) \leq ([\mathbf{W}_n]_+\mathbf{u}_{n-1} + [\mathbf{W}_n]_-\mathbf{l}_{n-1})[\mathbf{W}_{n-1}]_+ + ([\mathbf{W}_n]_+\mathbf{l}_{n-1} + [\mathbf{W}_n]_-\mathbf{u}_{n-1})[\mathbf{W}_{n-1}]_-.$$

These bounds are equivalent to Zhang et al. [59]'s Eq. (10) and (11) for obtaining $\tilde{\mathbf{L}}_{n-1}$ and $\tilde{\mathbf{U}}_{n-1}$, and thus $\hat{\mathbf{L}}_{n-1} = \tilde{\mathbf{L}}_{n-1}$ and $\hat{\mathbf{U}}_{n-1} = \tilde{\mathbf{U}}_{n-1}$.

**Equivalence on remaining layers**  Next, we use mathematical induction to show that $\hat{\mathbf{L}}_i = \tilde{\mathbf{L}}_i$ and $\hat{\mathbf{U}}_i = \tilde{\mathbf{U}}_i$ hold for $i = n-2, n-3, \cdots, 1$. Suppose we have shown that $\hat{\mathbf{L}}_i = \tilde{\mathbf{L}}_i$ and $\hat{\mathbf{U}}_i = \tilde{\mathbf{U}}_i$ hold for $k+1 \leq i \leq n$, and we aim to show that $\hat{\mathbf{L}}_k = \tilde{\mathbf{L}}_k$ and $\hat{\mathbf{U}}_k = \tilde{\mathbf{U}}_k$. We focus on the lower bound first, and the upper bound can be similarly proved. $\hat{\mathbf{L}}_k$ is the lower bound of

$\mathbf{J}_k(\mathbf{x}) = \mathbf{J}_{k+1}(\mathbf{x})\Delta_k(\mathbf{x})\mathbf{W}_k$ computed by bound propagation as

$$\mathbf{J}_{k+1}(\mathbf{x})\Delta_k(\mathbf{x})\mathbf{W}_k$$

$$\geq \sum_{\substack{([\Delta_k(\mathbf{x})]_{jj}=0 \text{ or } [\Delta_k(\mathbf{x})]_{jj}=1) \\ \text{and } ([\mathbf{L}_{k+1}]_j \geq 0 \text{ or } [\mathbf{U}_{k+1}]_j \leq 0)}} [\mathbf{J}_{k+1}(\mathbf{x})]_j([\Delta_k(\mathbf{x})]_{jj}[\mathbf{W}_k]_{j,:}) \tag{16}$$

$$+ \sum_{\substack{[\Delta_k(\mathbf{x})]_{jj}=[0,1], \\ [\mathbf{L}_{k+1}]_j \geq 0}} [\mathbf{J}_{k+1}(\mathbf{x})]_j[\mathbf{W}_k]_{j,:,-} \tag{17}$$

$$+ \sum_{\substack{[\Delta_k(\mathbf{x})]_{jj}=[0,1], \\ [\mathbf{U}_{k+1}]_j \leq 0}} [\mathbf{J}_{k+1}(\mathbf{x})]_j[\mathbf{W}_{n-1}]_{j,:,+} \tag{18}$$

$$+ \sum_{[\mathbf{L}_{k+1}]_j < 0 < [\mathbf{U}_{k+1}]_j} ([\mathbf{L}_{k+1}]_j[\mathbf{u}_i']_j[\mathbf{W}_k]_{j,:,+} + [\mathbf{U}_{k+1}]_j[\mathbf{u}_i']_j[\mathbf{W}_k]_{j,:,-}), \tag{19}$$

where Eq. (16) is a special case with fixed $[\Delta_k(\mathbf{x})]_{jj}$ and the sign of $[\mathbf{J}_{k+1}(\mathbf{x})]_j$ is also fixed, Eq. (17) and Eq. (18) are by Eq. (9), and Eq. (22) is by the interval relaxation in Eq. (15). By merging, Eq. (16), Eq. (17) and Eq. (18), the bound can be further simplified into

$$\mathbf{J}_{k+1}(\mathbf{x})\Delta_k(\mathbf{x})\mathbf{W}_k$$

$$\geq \sum_{[\mathbf{L}_{k+1}]_j \geq 0} [\mathbf{J}_{k+1}(\mathbf{x})]_j[\mathbf{u}_k']_j[\mathbf{W}_k]_{j,:,-} + [\mathbf{J}_{k+1}(\mathbf{x})]_j[\mathbf{l}_k']_j[\mathbf{W}_k]_{j,:,+} \tag{20}$$

$$\sum_{[\mathbf{U}_{k+1}]_j \leq 0} [\mathbf{J}_{k+1}(\mathbf{x})]_j[\mathbf{l}_k']_j[\mathbf{W}_k]_{j,:,-} + [\mathbf{J}_{k+1}(\mathbf{x})]_j[\mathbf{u}_k']_j[\mathbf{W}_k]_{j,:,+} \tag{21}$$

$$+ \sum_{[\mathbf{L}_{k+1}]_j < 0 < [\mathbf{U}_{k+1}]_j} ([\mathbf{L}_{k+1}]_j[\mathbf{u}_i']_j[\mathbf{W}_k]_{j,:,+} + [\mathbf{U}_{k+1}]_j[\mathbf{u}_i']_j[\mathbf{W}_k]_{j,:,-}). \tag{22}$$

Eq. (22) corresponds to Zhang et al. [59]'s "(14)". And when using linear bound propagation for cases with $[\mathbf{L}_{k+1}]_j \geq 0$ or $[\mathbf{U}_{k+1}]_j \leq 0$, bounds are propagated to $\mathbf{J}_{k+1}(\mathbf{x})$ and then the linear coefficients are merged with weights of the next layer $\mathbf{W}_{k+1}$, which corresponds to Zhang et al. [59]'s "(17)". Thus we have so far showed the equivalence in computing the lower bounds, and the equivalence can also be similarly derived for the upper bounds.