# OpenReview forum: "Efficiently Computing Local Lipschitz Constants of Neural Networks via Bound Propagation"
_NeurIPS.cc/2022/Conference — NeurIPS 2022 Accept_

### Official Review · Reviewer_LzFR · 2022-07-08

**Rating:** 6
**Confidence:** 3
**Soundness:** 3 good
**Presentation:** 3 good
**Contribution:** 2 fair

**Summary:**

The paper presents a novel method to compute upper bounds to the local Lipschitz constant of a neural network, by using linear bound propagation to bound the norm of the Clarke Jacobian of the network. The authors demonstrate that the proposed approach computes tighter bounds than the state-of-the-art in reasonable time on synthetic networks, and on networks trained for popular image datasets. Finally, the method is tested on monotonicity analysis, as an example application.

**Questions:**

- It would be interesting to compare the presented branching strategy against existing strategies proposed in the context of network verification, such as input splitting or FSB from https://arxiv.org/abs/2104.06718, which was notably applied in beta-CROWN

- Given that it relies on similar techniques, RecurJac should trivially support convolutional layers. Implementing this support would provide a fairer comparison.

- It would be interesting to see whether the baselines yield any different results in Table 4, for the monotonicity analysis.

**Limitations:**

The authors adequately discuss the limitations of their work. However, it would be better for the reader to have them condensed in a section in the appendix.

**Strengths And Weaknesses:**

The authors make a more careful use of linear bound propagation techniques in the context of estimating a network's local Lipschitz constant, resulting in tighter bounds than the state-of-the-art estimators in reasonable time. Furthermore, the algorithm supports branching to tighten the bounds, and a custom branching strategy is presented, which appears to work well in the considered experimental settings. While the approach is not entirely original, and it draws on existing techniques (such as RecurJac), it is clearly presented and effective. I believe it might be of interest to the community.

For what concerns the weaknesses, the experimental section is somewhat limited, as it mostly relies on small and synthetic networks. It would be interesting to test the algorithm on larger networks, and possibly on neural network verification, as for instance done by the RecurJac authors. At that point, I would be curious to see how Lipschitz-based algorithms compare with common incomplete verifiers (especially those based on optimized linear bounds such as alpha and beta-CROWN). Analogously, it would be interesting to see an extension of section 4.2 on l_2 norm (which bounding algorithms for Lipschitz constants routinely support).

---

> ### Author Response · Authors · 2022-08-02
> **Author response for Reviewer LzFR**
>
>
> We thank the reviewer for the constructive feedback and we have included more experimental evaluation in our response. One reason we could not conduct these experiments is because many existing methods (such as RecurJac) were only implemented for simple MLP networks. During the response period we re-implemented RecurJac to allow a fair comparison, and in addition we believe it is not fair to directly compare to many existing robustness verifiers, as we will discuss below.
>
> ## Comparisons on Larger networks (CIFAR-10 and TinyImageNet)
>
> We further added experiments to compare our method with RecurJac on larger models on  CIFAR-10 and TinyImageNet, by re-implementing RecurJac's relaxation in our code with efficient convolution support. Using same settings in Table 3, we compare the average local Lipschitz constants:
>
> | Dataset | Model | RecurJac | Ours |
> | --- | ----------- | ----------- | ----------- |
> | CIFAR-10 | CNN-2C2F | 50227.09 | 18638.14 |
> | CIFAR-10 | CNN-4C2F | 7735673.5 | 1049447.88 |
> | TinyImageNet | CNN-2C2F | 9548.22 | 4556.61 |
> | TinyImageNet | CNN-2C2F | 111570.94 | 25096.26 |
>
> Our method produces much tighter results than RecurJac due to the tighter relaxations.
>
>
> ## Comparisons on monotonicity analysis with new baselines
>
> For the monotonicity analysis, we add an comparison with RecurJac. RecurJac can also provide element-wise bounds for the Clarke Jacobian, which can be used for verifying monotonicty. Using settings from Table 4, we show results below, where our method verifies more monotonicity than RecurJac. We expect the gap to be larger when the dataset and models are larger.
>
> | Feature | $\uparrow$ (RecurJac) | $\downarrow$ (RecurJac) | $\uparrow$ (Ours) | $\downarrow$ (Ours) |
> | --- | ----------- | ----------- | ----------- | ---- |
> | Age |  32\% | 0\% | 40\% | 0\% |
> | Education | 55\% |  0\% | 58\% | 0\% |
> | Capital gain |  0\% | 0\% | 5\% | 0\% |
> | Capital loss |  4\% | 0\% | 7\% | 0\% |
> | Hours-per-week |  95\% | 0\% | 98\% | 0\% |
>
> ## Comparisons to neural network robustness verification
>
> We believe it is hard to conduct a fair comparison here. There are multiple kinds of neural network verification.
> The verification mentioned by the reviewer is usually for **robustness verification**, which aims to verify the output of the network given a perturbation on the input.
> However, in this paper, **we focus on verifying local Lipschitz constants, which is actually a different kind of neural network verification**. Instead of verifying the output of the network, it aims to **verify how fast the output changes when the input changes** (e.g., verification of Jacobians or derivatives).
>
> Robustness verification methods are *not* directly applicable to verifying local Lipschitz constants, and also cannot verify monotonicity (which requires bounds on Jacobian) as we did.
> On the other hand, we do not expect that using the local Lipschitz constants to *indirectly* verify robustness can outperform specialized robustness verification methods.
> Results in [RecurJac's slides (PDF page 32)](https://www.huan-zhang.com/pdf/RecurJac_Slides.pdf) showed that RecurJac could not outperform CROWN on $\ell_\infty$ robustness verification.
> If one's goal is to verify the robustness rather than Lipschitzness, we believe it is more suitable to directly use robustness verifiers.
>
>
> ## Extension to $\ell_2$-norm
>
> Our method may be used to produce $\ell\_2$ results by replacing $|\cdot|$ in Section 4.2 with $ (\cdot)^2 $ for $\ell\_2$ norm. But with the current bound propagation framework, $\ell\_2$ results tend to be loose. It is possibly because the current bound propagation considers *elementwise* linear bounds, which is more suitable for $\ell\_\infty$ norm but not $\ell\_2$ norm (this limitation applies to RecurJac as well). At least nontrivial efforts will be needed to produce tight $\ell\_2$ results (e.g., with nontrivial modification on the bound propagation), which are not considered in this paper.
>
> We revised our writing to clearly restrict our claims to the $\ell_\infty$ case, and we also added a sentence in the conclusion to mention this limitation and future work. Previous state-of-the-art works on $\ell\_\infty$ local Lipschitz constants, such as LipMIP and RecurJac, did not tightly handle the $\ell\_2$ case either, while LipSDP is designed for $\ell\_2$ but cannot be effectively applied to $\ell\_\infty$ and we have to loosely convert its $\ell\_2$ results into $\ell\_\infty$. Since most previous works focus on a single norm only, and the technique used to achieve the tightest bounds for different norms are likely to be different, we believe it is reasonable to focus on one specific norm here.

---

> > ### Author Response · Authors · 2022-08-02
> > **Author response for Reviewer LzFR  (cont.)**
> >
> > ## Branching Heuristic
> >
> > In neural network verification, input splitting is more suitable for tasks where the input dimension is small, but input dimensions are usually large on image tasks. Existing branching heuristics for splitting activations cannot be directly applied to our scenario here. It is because activations are on the forward graph, but we need to bound the Clarke Jacobian on the backward graph, and thus we need to consider the effect of splitting an activation from the forward graph on the bound computation for the backward graph. We propose a simple heuristic in Section 4.4.
> >
> > We tried adapting the filtering method from FSB to our branching heuristic, which did not produce better results on the MLP model for MNIST (412.48 by filtering among 3 candidates v.s. 397.25 with the heuristic in the paper only). This is probably because the filtering procedure causes additional time cost which makes the number of branches fewer given the limited time budget.  Considering more advanced branching heuristics will be possible extensions for this work.

---

> ### Author Response · Authors · 2022-08-08
> **Thank you for your constructive comments and we hope to hear from the reviewer before the discussion period is closed**
>
> Dear Reviewer LzFR,
>
> We thank you again for recognizing the importance of our work and giving us very helpful feedback. In our response, based on your suggestions, we **added experiments on larger models for CIFAR-10 and TinyImageNet**. We also **reimplemented RecurJac** in our codebase for a fair comparison and added experimental comparisons on CIFAR-10, TinyImageNet, and monotonicity analysis. We also discussed the questions on bounding $\ell_2$ norm Lipschitz constant, comparisons to robustness verifiers, and branching heuristics.
>
> We hope our response has addressed your questions and concerns. Since the discussion period is closing soon, we hope the reviewers can re-evaluate our paper based on our response. Thank you again and please let us know if you have any further questions.
>
> Sincerely,
> Paper 7279 Authors

---

> ### Comment · Reviewer_LzFR · 2022-08-09
> **Thank you for your thorough response**
>
> I thank the authors for their thorough response. It would be nice if the authors could include the discussion on the branching heuristic and to the l_2 norm extension to the paper, perhaps in the appendix: I believe this might be of interest to many readers.
> I increased my score by 1.

---

### Official Review · Reviewer_2723 · 2022-07-09

**Rating:** 4
**Confidence:** 4
**Soundness:** 2 fair
**Presentation:** 2 fair
**Contribution:** 2 fair

**Summary:**

This paper extends existing Clarke Jacobian estimation from interval bounds to the linear relaxation. The authors further apply the branch-and-bound method to tighten the estimation when the time budget allows. The experiments show that the linear-relaxed Clarke Jacobian with branch-and-bound produces more precise results than selected benchmarks. The authors also use the monotonicity analysis as an application of Clarke Jacobian.

**Questions:**

1. My experience with (interval)-Clarke Jacobian is that it provides very loose result in the $\ell_2$-case. The $\ell_2$ matrix norm (largest singular value) computation is also very different from $\ell_\infty$-norm. As a result, section 4.2 would be very different in the $\ell_2$-case. I do not know how much the improvement brought from linear bound propagation would be in the $\ell_2$-case. If this work only applies to $\ell_\infty$ and $\ell_1$ (linear constraint) perturbation space, please be upfront in the abstract or intro; otherwise, please provide some comparison with LipSDP in the $\ell_2$-case.
2. Optimality result: The optimality result of CROWN has been shown in [2]. The linear constraint propagation provided in this paper appears very similar to CROWN, except this work is for the backward Clarke Jacobian analysis. Why is Theorem 4 not a corollary of the theorem in [2], given Propositions 2 and 3?

**Limitations:**

No major concerns. One suggestion for paper writing: please use other graphical features than colors to distinguish different parts of the plot. This is not friendly for color-blind readers or black and white printers.

**Strengths And Weaknesses:**

Strengths:
This work provides tighter estimation of Clarke Jacobian norm estimation, which has practical benefits for the local Lipschitz constant evaluation. Local Lipschitz constant is an essential mathematical property of the neural network in an input region.

Weaknesses:
1. This work is incremental. Tightening neural-network certification from interval bounds to linear relaxation has been studied in a number of works, for example, [1] and [2]. The branch-and-bound method has also been used to compute the Lipschitz constant of neural networks [3]. This work brings little new insight or understanding to neural-network certification. The benefit is mostly practical from compiling known methods together, which is not surprising.
2. Line 179 mentions that in this paper, the authors mainly consider $\ell_\infty$-norm *for simplicity*. I doubt whether this is *for simplicity* or linear relaxed Clarke Jacobian is only able to handle linearly constraint perturbation space in practice (see the Questions section below).
3. The evaluation has doubts. The first part is related to 2. LipSDP is designed to compute Lipschitz constant with respect to $\ell_2$ perturbations. Scaling the result of LipSDP for $\ell_2$ perturbations seems an unfair comparison to LipSDP, unless the authors make it clear this work does not apply to $\ell_2$ perturbations. The second part is for the monotonicity analysis. The authors provide some numbers from experiments, but there is no justification on how good the result is.

[1]Zhang, H., Weng, T., Chen, P., Hsieh, C., and Daniel, L. Efficient neural network robustness certification with general activation functions. In Advances in Neural Information Processing Systems, pp. 4944–4953, 2018. URL https://proceedings.neurips.cc/paper/2018/hash/d04863f100d59b3eb688a11f95b0ae60-Abstract.html.

[2]Lyu, Z., Ko, C., Kong, Z., Wong, N., Lin, D., and Daniel, L. Fastened CROWN: tightened neural network robustness certificates. In The Thirty-Fourth AAAI Conference on Artificial Intelligence, AAAI 2020, The Thirty-Second Innovative Applications of Artificial Intelligence Conference, IAAI 2020, The Tenth AAAI Symposium on Educational Advances in Artificial Intelligence, EAAI 2020, New York, NY, USA, February 7-12, 2020, pp. 5037–5044, 2020. URL https://aaai.org/ojs/index.php/AAAI/article/view/5944.

[3]Bhowmick, A., D’Souza, M., and Raghavan, G. S. Lipbab: computing exact lipschitz constant of relu networks. In International Conference on Artificial Neural Networks, pp. 151–162. 332 Springer, 2021.

---

> ### Author Response · Authors · 2022-08-02
> **Author response for Reviewer 2723**
>
>
> We thank the reviewer for the constructive feedback. As suggested by the reviewer, we revised our paper and made the claims more clear in our paper that we focus on $\ell_\infty$ norm, and also explained the difference between our work and previous works on tightening neural network verification, as well as adding more baselines on monotonicity analysis. We hope the reviewer can re-evaluate our paper based on our response below:
>
> ## Comparison with previous works on tightening certification and optimality results
>
> We believe tightening certification in [1, 2] is significantly different from the linear relaxation problem in bounding Clarke Jacobian. **[1] and [2] both focused on the linear relaxation for simple activation functions** such as $\sigma(x)$ for pre-activation value $x$, where $\sigma(x)$ is solely a function of $x$.
> However, in this paper we want to bound $[\mathbf{J}\_{i+1}(\mathbf{x})]\_j [\Delta\_i(\mathbf{x})]\_{jj}$ which is not a single function of $[\mathbf{J}\_{i+1}(\mathbf{x})]\_j$, but **a group of functions of $[\mathbf{J}\_{i+1}(\mathbf{x})]\_j$ given different $[\Delta\_i(\mathbf{x})]\_{jj}$
> (violet area in Figure 4)**.
>
> Specifically, [1] proposed a heuristic to adaptively choose a lower bound relaxation for ReLU, which has no guarantee on the optimality. [2] proposed an optimization-based approach to choose optimal relaxations for activations.
> In contrast, we bound **a group of functions** in Clarke Jacobian computation, instead of a single activation function (thus the optimality result in [2] is *not directly applicable* here). **Our linear relaxation is optimal (proved in Theorem 4), and importantly, it is closed-form and does not need gradient-based optimization unlike [2].**
>
>
> ## Limitations on $\ell_\infty$-norm
>
> - First, we apologize for the misleading wording in our paper, and we revised our writing to clearly restrict our claims to the $\ell_\infty$ case, avoided saying "for simplicity". We also added a sentence in the conclusion to mention this limitation and future work. The modification has been highlighted in blue in the paper.
>
> - Second, our method can be used to produce $\ell\_2$ results by replacing $|\cdot|$ in Section 4.2 with $ (\cdot)^2 $ for $\ell\_2$ norm, although we agree with reviewer that with the current bound propagation framework, $\ell\_2$ results tend to be loose. It is possibly because the current bound propagation considers *element-wise* linear bounds, which is more suitable for $\ell\_\infty$ norm but not $\ell\_2$ norm. Tightening the $\ell\_2$ norm results will require new techniques (e.g., with nontrivial modification on the bound propagation), which are not considered in this paper.
>
> ## Evaluation of $\ell_2$-norm baselines such as LipSDP has doubts
>
> We apologize for not being clear on the comparison to LipSDP. We have revised our paper and noted that the results from LipSDP are not from the same setting. We do agree that it is not a completely fair comparison since LipSDP is designed for $\ell\_2$ norm, and we just wanted to include as many baselines as possible for reference, while other baselines such as LipMIP and RecurJac focus on $\ell\_\infty$ norm.  We believe it is reasonable to focus on one specific norm only here, and it will be interesting for future work to consider other norms.
>
>
> ## Baselines on the monotonicity analysis
>
> For the monotonicity analysis, we add an comparision with RecurJac. RecurJac can also provide element-wise bounds for the Clarke Jacobian, which can be used for verifying monotonicty. Using settings from Table 4, we show results below, where our method verifies more monotonicity than RecurJac (a higher number is better). We expect the gap to be larger when the dataset and models are larger.
>
> | Feature | $\uparrow$ (RecurJac) | $\downarrow$ (RecurJac) | $\uparrow$ (Ours) | $\downarrow$ (Ours) |
> | --- | ----------- | ----------- | ----------- | ---- |
> | Age |  32\% | 0\% | 40\% | 0\% |
> | Education | 55\% |  0\% | 58\% | 0\% |
> | Capital gain |  0\% | 0\% | 5\% | 0\% |
> | Capital loss |  4\% | 0\% | 7\% | 0\% |
> | Hours-per-week |  95\% | 0\% | 98\% | 0\% |
>
>
> ## Accessibility of Figures
>
> We thank the reviewer for the suggestion and will update the figures to improve the accessibility.
>
>
>
> (The deleted post below was a duplicate accidentally created during submission.)

---

> > ### Comment · Reviewer_2723 · 2022-08-06
> > **Post-rebuttal response**
> >
> > I have carefully read the revised paper and the reponses. The authors have acknowledged that this work only applies to the $\ell_\infty$-perturbations.
> > > our method can be used to produce $\ell_2$  results by replacing $|\cdot|$ in Section 4.2 with $(\cdot)^2$ for $\ell_2$ norm.
> >
> > $(\cdot)^2$ in the multidimensional case is equivalent to the Frobenius norm, which is theoretically known larger than $\ell_2$ operator-norm. I do not think this is a valid argument for the empirical work. The goal is to give a precise upper bound. We can always find a trivial loose upper bound, so I do not think this is a satisfying response.
> >
> > The major contribution of this work is to use linear bound instead of interval bound to compute the Clarke Jacobian w.r.t. $\ell_\infty$-perturbations. This backward Jacobian computation for $\ell_\infty$-perturbations is not very different from CROWN's setting, so the novelty in this work is minimal, especially given that there have been quite a few works following CROWN using linear bound to tighten the propagation result. As a result, it is subpar for NeurIPS standard.
> >
> > Also, I think that the authors should credit previous CROWN-like works that have already proved the optimality of the linear bound propagation for their settings.

---

> > > ### Author Response · Authors · 2022-08-08
> > > **Follow-up response (2/2)**
> > >
> > >
> > > > especially given that there have been quite a few works following CROWN using linear bound to tighten the propagation result
> > >
> > > We thank the reviewer again for pointing out these paper and we have cited them. We have revised the paper to mention the related papers after Theorem 4 and discussed the difference.
> > > As we mentioned in the initial reply, **improvement in those works is not applicable to the settings here, and our proposed method is also different which achieves the provable optimality without gradient-based optimization**.
> > > Speficially, methods in those works are not applicable to the Clarke gradient due to different nonlinearities;
> > > in addition, Lyu et al., 2020 defined a tightest bounding line as `no tighter bounding line than itself` and there can be many different tightest bounding lines that need to be chosen by an optimization method, while **our linear relaxation is closed-form and tighter than any other linear relaxation and is the unique tightest one**.
> > > We believe it is unfair to judge the novelty of our paper by referring to papers which have different scopes not applicable to our settings, and also different methods and results.

---

> > > > ### Comment · Reviewer_2723 · 2022-08-08
> > > > **increase my score**
> > > >
> > > > This work is mostly practical and has limited theoretical or technical novelty: I do not find the differences pointed out by the authors fundamentally restricting linear bounding, for example, appearing impossible on the surface; and in this work, the application of linear bounding is still straightforward.
> > > >
> > > > Previously, the paper made false claims on its application scope, i.e., working for $\ell_p$-perturbations. The revised paper has already restricted the application scope to $\ell_\infty$-perturbations only. As a result, I increased my score by 1.

---

> > > ### Author Response · Authors · 2022-08-08
> > > **Follow-up response (1/2)**
> > >
> > >
> > > We thank the reviewer for the prompt response. While the comments on $\ell_2$ norm seems to be a misunderstanding of our comments and previous baselines, we greatly appreciate the criticism from the reviewer since it is very helpful for us to rethink our contribution and novelty. We hope our response below will clarify misunderstandings and make the contributions of our work more clear:
> > >
> > > ## Limitations on $\ell_2$ norm
> > >
> > > We respectfully disagree with the reviewer's comments on handling $\ell_2$ norm from three perspectives:
> > >
> > > - **None of previous state-of-the-art methods including LipMIP, LiPSDP, RecurJac could tightly handle both $\ell_2$ and $\ell_\infty$ simultaneously**. Typically, bounding different norm requires different techniques. We believe it is unfair to require us to tighten both kinds of bounds.
> > >
> > > - We have **clearly discussed this limitation** in our response and our paper: immediately following the sentence you quoted, we clear stated `we agree with reviewer that with the current bound propagation framework, results tend to be loose`. We feel rejecting our paper based on this  limitation is unfair since we have clearly discussed our limitations on $\ell_2$ norm and none of existing works can effectively tighten both norms.
> > >
> > > - Regarding your particular comments on Frobenius norm, $\ell\_2$ operator norm and Frobenius norm are the same when considering one particular output at each time (e.g., when one target class is considered). In these cases, the output dimension of the function is 1 and the Jacobian is a row-vector.
> > >
> > > ## Novelty and Contributions
> > >
> > > >This backward Jacobian computation for $\ell_\infty$-perturbations is not very different from CROWN's setting, so the novelty in this work is minimal
> > >
> > > Although this work is based on CROWN-like linear bound propagation principle, we believe this does not restrict the contribution of our paper, and our formulation of the linear bound propagation in the setting of bounding local Lipschitz constant is actually unique and beneficial, as we discuss below:
> > >
> > > - We did not naively apply CROWN directly  and we made novel technical contributions to handle the bounds on Jacobian, including bounding a group of activation functions as we explained in the initial reply "Comparison with previous works on tightening certification and optimality results". Our novel treatment of these non-linearities allows us to achieve better relaxations and tighter bounds.
> > >
> > >
> > > - It is an important contribution that we reformulate the problem of bounding local Lipschitz constant under the linear bound propagation framework, which has not been demonstrated before.
> > > Existing algorithms (such as LipMIP and Recurjac) are very specialized to computing Lipschitz constant.
> > > Our formulation of the problem in a more general linear bound propagation framework not only allows us to **compute a tighter bound** as discussed above,
> > > but also **makes the existing highly optimized toolbox on linear bound propagation available to the new problem** of bounding Lipschitz constant, which was not possible before.
> > > Unlike existing specialized algorithms, our algorithm can be incorporated into a high quality linear bound propagation toolbox, which enables us to utilize efficient convolution support for linear bound propagation and conduct branch-and-bound to allow for an iterative improvements of the bounds.
> > > This allows the entire ML community to enjoy the state-of-the-art progress on tightening local Lipschitz and Jacobian bounds.
> > >
> > >
> > > - Experimentally, our paper enables the computation of tighter $\ell_\infty$ Lipschitz constants on a broader class of functions (e.g., CNNs) as well as larger networks for larger datasets, which was never demonstrated before, thanks to our formulation of computing local Lipschitz constant under the linear bound propagation framework, which can be efficiently computed on a GPU accelerator. This is in a clear contrast with many existing approaches such as LipMIP and LipSDP that require expensive solvers.
> > >
> > > Overall, our work clearly pushes the boundary of existing state-of-the-art research on tightening Jacobian bounds and $\ell_\infty$ norm local Lipschitz constants, which could not be achieved by using CROWN alone nor existing works on bounding local Lipschitz constants. This is **clearly beyond the rating of 3**: `a paper with technical flaws, weak evaluation, inadequate reproducibility and incompletely addressed ethical considerations`. We hope the reviewer can reconsider these contributions of our paper based on our response.

---

### Official Review · Reviewer_jXRD · 2022-07-10

**Rating:** 7
**Confidence:** 4
**Soundness:** 3 good
**Presentation:** 3 good
**Contribution:** 3 good

**Summary:**

This work presents a method to efficiently compute upper bounds for the Local Lipschitz constant of ReLU networks. In short, Interval Bound Propagation techniques are applied to the backward computational graph, which yields an upper bound on the norm of the
Clarke Jacobian of the network, which constitutes an upper bound for the Local Lipschitz constant. Careful consideration of the bounds for well-known activation functions provide tighter results over previous baselines. Such improvements are confirmed on experiments using MLPs and Convolutional networks trained on MNIST/CIFAR10/TinyImagenet.

*** after rebuttal ***
Authors have addressed my main concerns. I am inclined to increase my score.

**Questions:**

1. Isn't RecurJac precisely an application of Interval Bound Propagation on the backward computational graph of the network (the Jacobian)? Could you expand and clarify the differences between your approach an RecurJac?
2. Could you clarify which "Chain rule" are you using for composition of non-differentiable functions?
3. Do you have new experimental results with confidence intervals, that you could add to the tables?

**Limitations:**

No negative societal impact. The main limitation is the cost of Branch-and-bound which the authors somehow address by providing an option to use the method without BaB.

**Strengths And Weaknesses:**

**Originality**: I think the main weakness is in this front, as to me the contribution is not clear. The authors claim that their contribution is to "generalize existing bound propagation methods originally used for NN verification, to a higher order backward computational graph for bounding the Clarke Jacobian". However, to me it appears that [B] precisely does that, with the caveat that their results are for differentiable activation functions, when the notion of Clarke Jacobian is just the normal Jacobian function. As such, I believe that this work rather adapts the technique in [B] to the setting of non-differentiable activation functions like ReLU, together with a more careful approach that provides tighter bounds in some cases, compared to the methods in [B]. This should be clarified by the authors and if it is the case, the claims should be toned down. In summary, I believe that applying Interval Bound Propagation techniques to the Jacobian (which is the backward computational graph) is already done in [B], although in a slightly different setting.

**Quality and Clarity**: The paper is in general well-written and easy to follow, and the bibliography appears all of previous related work. The differences with most previous works are clearly explained, again with the exception of RecurJac [B], where I believe the authors should expand. Also in line 151 the authors claim "Chain rule can be used to compute the Clarke Jacobian for the whole ReLU network", however it is well known that the notion of "Chain rule" is more complicated when dealing with non-smooth compositions, I would like the authors to clarify which "Chain rule" are they talking about. See for example Theorem 4 in [A].

**Significance**: I believe it is important to extend the method in [B] to the case of non-smooth activations, as activations like ReLU and LeakyReLU are commonly used in practice despite the fact that they do not have a "normal" gradient at all points. Moreover, **Theorem 4** in this work is a tightness result which implies that the relaxation presented in this work is the tightest possible, which is an important step in this line of research. However, related to my main criticism of the work, if indeed the idea of applying interval bound propagation to the backward computational graph is already present in [B], the method presented here would be more of a refinement of a previous method rather than a completely different approach, which is less significant.

In the experiments it is not clear why RecurJac is used despite it being tailored for differentiable activation functions, can the authors clarify if using RecurJac for non-smooth activation functions still provide a valid upper bound for the norm of the Jacobian? Another issue with the experiments is a lack of confidence intervals. Because the networks used are a result of a stochastic process (SGD) there might be differences due to randomness, and without confidence intervals it is hard to assess if the improvement is significant. For example the method LipBAB achieves a runtime of 2.92 for MLPs of width 32 and 4 layers, which is much faster than the method presented here, while providing a Lipschitz constant as good. However in other cases it is considerably slower and provides worse bounds. Adding confidence intervals could possibly let the authors conclude if their method is **consistently outperforming the baselines**.

Nevertheless,their method (without Branch-and-bound) appears to be consistently faster and more scalable than RecurJac.

**References**
[A] Support functions of Clarke's generalized jacobian and of its plenary hull. Cyril Imbert.
[B] RecurJac: An Efficient Recursive Algorithm for Bounding Jacobian Matrix of Neural Networks and Its Applications, Huan Zhang, Pengchuan Zhang, Cho-Jui Hsieh, AAAI 2019

---

> ### Author Response · Authors · 2022-08-02
> **Author response for Reviewer jXRD**
>
>
> We thank the reviewer for the detailed, encouraging and helpful feedback. We feel the questions related to our contributions are largely due to misunderstandings. We made clarifications and also provided additional results on repeated experiments below.
>
> ## Our contributions, originality and relations to RecurJac
>
> The main difference of our work is on the **linear** bound propagation approach we used, not on the non-smooth activation function, as we elaborate below:
>
> - First of all, **our method is not based on interval bound propagation, but bound propagation with linear relaxations**.
> The difference is that *interval bounds* use constant bounds, while *linear relaxation* uses linear functions as bounds which can eventually produce tighter bounds. Figure 4 in the paper illustrates the difference, where interval bounds (dashed green lines) are horizontal lines, while our linear relaxation (dashed blue and red lines) has non-zero slopes and is the tightest linear relaxation (Theorem 4). We have edited the paper to emphasize as "we generalize existing **linear** bound propagation methods".
>
> - Second, while RecurJac uses bound propagation, **it essentially uses interval bound propagation in nontrivial cases** (when both ReLU and the sign of Clarke Jacobian are unstable), as shown in Figure 4. Thus **RecurJac is not able to generally perform linear bound propagation, since it does not have a linear relaxation in nontrivial cases**. In contrast, we use tight linear relaxation and handle linear bound propagation on the backward computational graph. We have revised the introduction to make the comparison more clear (the modification is highlighted in blue). Our method produces significantly tighter results than RecurJac. In addition, we also support branch-and-bound to further tighten the results.
>
> - Third, the main difference between our work and RecurJac is *not* on whether the activation is smooth. RecurJac fully supports non-smooth activation functions and their paper included experiments on (leaky-)ReLU networks. Our work is not an extension in this regard. ReLU activation $\sigma(x)$ is non-smooth only at $x=0$, and its Clarke gradient at $x=0$ is $\partial \sigma(x)=[0,1]$. When bounding $\partial\sigma(x)$ given $l\leq x\leq u$, a special treatment is needed when exactly $l=0$ or $u=0$, and in these cases $\partial\sigma(x)\in [0,1]$. In this regard, we found a minor glitch in the [code of RecurJac](https://github.com/huanzhang12/RecurJac-and-CROWN/blob/master/bound_recurjac.py#L52) that the $l=0$ or $u=0$ case was not correct handled (they took $\partial\sigma(x)=0$ when $u\leq 0$ and $\partial\sigma(x)=1$ when $l\geq 0$, instead of taking $\partial\sigma(x)=[0,1]$). Although the issue did not affect the empirical results as it is rare to exactly have $l=0$ or $u=0$, our work is more rigorous with the formulation of Clarke Jacobian for non-smooth functions.
>
> - Additionally, by considering bound propagation on a computational graph in bounding the Clarke Jacobian (illustrated in Figure 3), we do not need to hand-derive the bound propagation as done in RecurJac paper and we can benefit from existing linear bound propagation frameworks with GPU acceleration, further improving the scalability of our approach.
>
> We thank the reviewer again for these very insightful questions and we will add these discussions in the final revision of our paper, where one additional page is allowed.

---

> > ### Author Response · Authors · 2022-08-02
> > **Author response for Reviewer jXRD (cont.)**
> >
> >
> > ## Confidence intervals for assessing improvments
> >
> > Our focus is on **scaling to relatively larger models** that previous methods either cannot handle or produce loose bounds, while **the smallest models in Table 1 are mostly for correctness checking** (i.e., our results should be no smaller than those by LipMIP which provides exact results on small models) and completeness of experiments. Therefore, for the smallest models such as the one mentioned by the reviewer, we do not expect to beat baselines with a large gap.
> >
> > On larger models, we add an experiment on MNIST with confidence interval as suggested. We train the same model 5 times with different random initialization (other settings are the same as those in Table 2), and compute local Lipschitz constant for each model. We report the mean and standard deviation for the 5 runs in each setting below:
> >
> > **3-layer MLP**
> >
> > | Method | Value | Runtime |
> > | --- | --- | --- |
> > | NaiveUB | 3218.17 $\pm$ 289.99 | 0.00 $\pm$ 0.00 |
> > | LipMIP | 14053.27 $\pm$ 264.03 | 120.29 $\pm$ 0.11 |
> > | LipBaB | 988.66 $\pm$ 86.93 | 63.01 $\pm$ 0.63 |
> > | RecurJac | 1153.68 $\pm$ 97.87 | 0.38 $\pm$ 0.08 |
> > | Ours (w/o BaB) | **734.40 $\pm$ 63.69** | 4.26 $\pm$ 0.37 |
> > | Ours (w/ BaB) | **423.30 $\pm$ 38.25** | 58.54 $\pm$ 3.15 |
> >
> > **CNN-2C2F**
> >
> > | Method | Value | Runtime |
> > | --- | --- | --- |
> > | NaiveUB | 92108.52 $\pm$ 13962.82 | 0.00 $\pm$ 0.00 |
> > | RecurJac | 11174.06 $\pm$ 1342.49 | 114.18 $\pm$ 3.40 |
> > | Ours (w/o BaB) | **4895.50 $\pm$ 517.39** | 7.89 $\pm$ 0.20 |
> > | Ours (w/ BaB) | **4878.96 $\pm$ 518.49** | 60.05 $\pm$ 0.01 |
> >
> > The results show that our methods **consistently and significantly outperform** the baselines for these MNIST models.
> >
> > In our response to Reviewer LzFR (see "Comparisons on Larger networks (CIFAR-10 and TinyImageNet)"), we show comparison on CIFAR-10 and TinyImageNet,  where we re-implemented RecurJac's bound propagation into our code which has efficient convolution support. Our method outperforms RecurJac with large margins.
> >
> >
> > ## Chain rules for Clarke Jacobian
> >
> > We follow the Chain rule for ReLU networks in the LipMIP paper (Jordan & Dimakis, 2020), and their definition matches Theorem 4 in [A] cited by the reviewer, except that we do not need to explicitly take the convex hull which is automatically satisfied here (the correctness is shown by Theorem 2 in Jordan & Dimakis, 2020).
> > We updated our writing in Section 4.1 to make the definition more clear and cited Jordan & Dimakis, 2020 and [A] there (the modification has been highlighted in blue).
> >
> > We hope the added experiments are helpful and the reviewer is clear on our contributions and the difference compared to RecurJac now. Please feel free to ask us any additional questions for clarification, thank you.

---

### Meta-Review · Area_Chair_3hcY · 2022-08-30

**Recommendation:** Accept
**Confidence:** Certain

**Metareview:**

The paper develops a methodology for computing the Lipschitz constant of ReLu neural networks (when the input perturbations are measured in the L_infinity norm). The method is based on computing tight upper bounds on the Clarke Jacobian. The basic idea is to apply Interval Bound Propagation techniques to the backward computational graph, which yields an upper bound on the norm of the Clarke Jacobian of the network. Experimental results show superiority over SOTA in terms of scalability, runtime, and the computed bound.

The reviewers had a number of concerns most of which were addressed during the discussion phase. I recommend that the authors revise the paper and the experiments according to the reviewers' comments as well as their own responses. The paper was also discussed among the reviewers. One main point of discussion was the novelty of the work (compared to prior art, e.g. Lyu et al) and the fact that bounding the norm of Clarke Jacobian seems to be only beneficial for L_infinity perturbations. However, some reviewers argued (and I agree) that the paper improves over SOTA methods quite notably in terms of efficiency and tightness, and the method scales to much larger models compared to prior works (scale is actually an important challenge in this topic). As a result, I would vote for accepting the paper.

As a matter of taste, I don't think I agree with this sentence in the abstract (and similar sentences in other parts of the paper): "Existing methods for computing Lipschitz constants either are  computationally inefficient or produce loose upper bounds." We do have good methods that provide non-trivial upper bounds on the Lipschitz constant of NNs. While I agree that the scalability of those methods are still to be improved, we can not really call them inefficient. Hence, I believe that this sentence (and similar sentences in the paper) could be better rephrased.

**Award:**

No

---

### Decision · Program_Chairs · 2022-09-14

Accept